# TAILORING MIXUP TO DATA USING KERNEL WARPING FUNCTIONS

## ABSTRACT

Data augmentation is an essential building block for learning efficient deep learning models. Among all augmentation techniques proposed so far, linear interpolation of training data points, also called *mixup*, has found to be effective for a large panel of applications. While the majority of works have focused on selecting the right points to mix, or applying complex non-linear interpolation, we are interested in mixing similar points more frequently and strongly than less similar ones. To this end, we propose to dynamically change the underlying distribution of interpolation coefficients through warping functions, depending on the similarity between data points to combine. We define an efficient and flexible framework to do so without losing in diversity. We provide extensive experiments for classification and regression tasks, showing that our proposed method improves both performance and calibration of models.

## 1 INTRODUCTION

The *Vicinal Risk Minimization (VRM)* principle (Chapelle et al., 2000) improves over the well-known *Empirical Risk Minimization (ERM)* (Vapnik, 1998) for training deep neural networks by drawing virtual samples from a vicinity around true training data. This data augmentation principle is known to improve the generalization ability of deep neural networks when the number of observed data is small compared to the task complexity. In practice, the method of choice to implement it relies on handcrafted procedures to mimic natural perturbations (Yaeger et al., 1996; Ha & Bunke, 1997; Simard et al., 2002). However, one counterintuitive but effective and less application-specific approach for generating synthetic data is through interpolation, or mixing, of two or more training data.

The process of interpolating between data have been discussed multiple times before (Chawla et al., 2002; Wang et al., 2017; Inoue, 2018; Tokozume et al., 2018), but *mixup* (Zhang et al., 2018) represents the most popular implementation and continues to be studied in recent works (Pinto et al., 2022; Liu et al., 2022b; Wang et al., 2023). Ever since its introduction, it has been a widely studied data augmentation technique spanning applications to *image classification and generation* (Zhang et al., 2018), *semantic segmentation* (Franchi et al., 2021; Islam et al., 2023), *natural language processing* (Verma et al., 2019), *speech processing* (Meng et al., 2021), *time series and tabular regression* (Yao et al., 2022a) or *geometric deep learning* (Kan et al., 2023), to that extent of being now an integral component of competitive state-of-the-art training settings (Wightman et al., 2021). The idea behind *mixup* can be seen as an efficient approximation of *VRM* (Chapelle et al., 2000), by using a linear interpolation of data points selected from within the same batch to reduce computation overheads.

The process of mixup as a data augmentation during training can be roughly separated in three phases: (i) selecting tuples (most often pairs) of points to mix together, (ii) sampling coefficients that will govern the interpolation to generate synthetic points, (iii) applying a specific interpolation procedure between the points weighted by the coefficients sampled. Methods in the literature have mainly focused on the first and third phases, *i.e.* the *process of sampling points to mix* through predefined criteria (Hwang et al., 2022; Yao et al., 2022a;b; Palakkadavath et al., 2022; Teney et al., 2023) and on the *interpolation itself*, by applying sophisticated and application-specific functions (Yun et al., 2019; Franchi et al., 2021; Venkataramanan et al., 2022; Kan et al., 2023). On the other hand, these interpolation coefficients, when they exist, are always sampled from *the same distribution* throughout training. Recent works have shown that mixing different points can result in arbitrarily incorrect

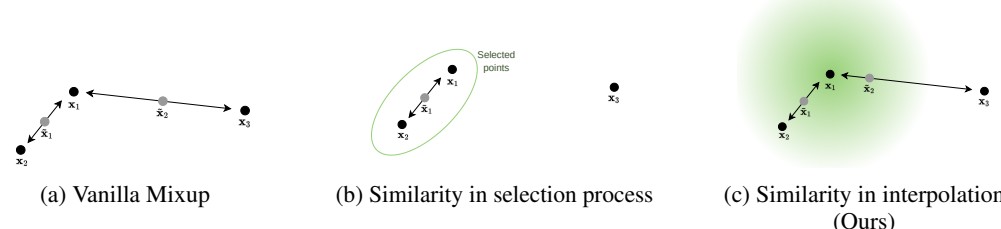

(a) Vanilla Mixup        (b) Similarity in selection process        (c) Similarity in interpolation (Ours)

Figure 1: Different approaches to take into account similarity between points in Mixup. *(Left)* Vanilla mixup process, any pair of points can be mixed with the same interpolation. *(Middle)* Only similar pairs are mixed. This restricts the possible synthetic data generated. *(Right)* We argue that similarity of points to mix should govern the interpolation coefficients, to avoid restricting possible directions of mixing.

labels especially in regression tasks (Yao et al., 2022a), while mixing similar points helps in diversity (Chawla et al., 2002; Dablain et al., 2022). Furthermore, several previous work have highlighted a trade-off between performance and calibration in Mixup (Thulasidasan et al., 2019; Pinto et al., 2022; Wang et al., 2023). Our main assumption is that mixing distant points can result in *manifold intrusion* (Guo et al., 2019; Baena et al., 2022), *i.e.* conflicts between the synthetic labels of the mixed samples and the labels of the original data, which can degrade calibration. However, the actual similarity between points is only considered through the *selection process*, and consequently these approaches generally suffer from three downsides: (i) they are inefficient, since the data used to mix are sampled from the full training set leading to memory constraints, and the sampling rates have to be computed beforehand; (ii) they reduce diversity in the generated data by restricting the pairs allowed to be mixed; (iii) it is difficult to apply the same approach between different tasks, such as classification and regression. In this work, we aim to provide an *efficient* and *flexible* framework for *taking similarity into account* when interpolating points without losing in *diversity*. Notably, we argue that similarity should influence the *interpolation coefficients* rather than the *selection process*. A high similarity should result in strong interpolation, while a low similarity should lead to almost no changes. Consequently, controlling the interpolation through the distance of the points to mix should improve calibration by reducing manifold intrusion and label noise. Figure 1 illustrates the different ways to take into account similarity between points in Mixup.

Our contributions [1] towards this goal are the following:

- We define *warping functions* to change the underlying distributions used for sampling *interpolation coefficients*. This defines a general framework that allows to disentangle inputs and labels when mixing, and spans several variants of mixup.

- We propose to then apply a *similarity kernel* that takes into account the distance between points to select a parameter for the warping function tailored to each pair of points to mix, governing its shape and strength. This tailored function warps the interpolation coefficients to make them stronger for similar points and weaker otherwise.

- We show that our *Kernel Warping Mixup* is general enough to be applied in classification as well as regression tasks, improves both *performance* and *calibration* while being an efficient data augmentation procedure. Our method is competitive with state-of-the-art approach and requires fewer computations.

## 2    RELATED WORK

In this section, we discuss related work regarding data augmentation through mixing data and the impact on calibration of modern neural network.

---

[1]Partial code is available as supplementary materials, full code will be released upon acceptance.

## 2.1 Data augmentation based on mixing data

The idea of mixing two or more training data points to generate additional synthetic ones has been developed in various ways in the literature.

**Offline interpolation** Generating new samples *offline*, *i.e.* before training, through interpolation of existing ones, is mainly used for *oversampling* in the *imbalanced setting*. Algorithms based on SMOTE (Chawla et al., 2002), and its improvements (Han et al., 2005; He et al., 2008; Dablain et al., 2022), are interpolating nearest neighbors in a latent space for minority classes. These methods are focusing on creating synthetic data for specific classes to fix imbalanced issues, and thus only consider interpolating elements from the same class.

**Online non-linear interpolation** Non-linear combinations are mainly studied for dealing with image data. Instead of a naive linear interpolation between two images, the augmentation process is done using more complex non-linear functions, such as cropping, patching and pasting images together (Takahashi et al., 2019; Summers & Dinneen, 2019; Yun et al., 2019; Kim et al., 2020) or through subnetworks (Ramé et al., 2021; Liu et al., 2022b; Venkataramanan et al., 2022). Not only are these non-linear operations focused on images, but they generally introduce a significant computational overhead compared to the simpler linear one (Zhu et al., 2020). The recent *R-Mixup* (Kan et al., 2023), on the other hand, considers other Riemannian geodesics rather than the Euclidean straight line for graphs, but is also computationally expensive.

**Online linear interpolation** Mixing samples online through linear interpolation represents the most efficient technique compared to the ones presented above (Zhang et al., 2018; Inoue, 2018; Tokozume et al., 2018). Among these different approaches, combining data from the same batch also avoids additional samplings. Several follow-up works extend mixup from different perspectives. Notably, *Manifold Mixup* (Verma et al., 2019) interpolates data in the feature space, *k-Mixup* (Greenewald et al., 2021) extends the interpolation to use *k* points instead of a pair, Guo et al. (2019) and Baena et al. (2022) apply constraints on the interpolation to avoid *manifold intrusion*, *Remix* (Chou et al., 2020) separates the interpolation in the label space and the input space and *RegMixup* (Pinto et al., 2022) considers mixup as a *regularization term*.

**Selecting points** A family of methods apply an online linear combination on *selected pairs* of examples (Yao et al., 2022a;b; Hwang et al., 2022; Palakkadavath et al., 2022; Teney et al., 2023), across classes (Yao et al., 2022b) or across domains (Yao et al., 2022b; Palakkadavath et al., 2022; Tian et al., 2023). These methods achieve impressive results on distribution shift and Out Of Distribution (OOD) generalization (Yao et al., 2022b), but recent theoretical developments have shown that much of the improvements are linked to a resampling effect from the restrictions in the selection process, and are unrelated to the mixing operation (Teney et al., 2023). These selective criteria also induce high computational overhead. One related approach is *C-Mixup* (Yao et al., 2022a), that fits a Gaussian kernel on the labels distance between points in regression tasks. Then points to mix together are sampled from the full training set according to the learned Gaussian density. However, the Gaussian kernel is computed on all the data *before* training, which is difficult when there is a lot of data and no explicit distance between them.

## 2.2 Calibration in classification and regression

*Calibration* is a metric to quantify uncertainty, measuring the difference between a model's confidence in its predictions and the actual probability of those predictions being correct.

**In classification** Modern deep neural network for image classification are now known to be *overconfident* leading to *miscalibration* (Guo et al., 2017). One can rely on *temperature scaling* (Guo et al., 2017) to improve calibration *post-hoc*, or using different techniques during learning such as *ensemble* (Lakshminarayanan et al., 2017; Wen et al., 2021), different losses (Chung et al., 2021; Moon et al., 2020), or through *mixup* (Thulasidasan et al., 2019). The problem of the trade-off between performance and calibration with Mixup have been extensively studied in previous work (Thulasidasan et al., 2019; Zhang et al., 2022; Pinto et al., 2022; Wang et al., 2023) However, Wang et al. (2023) recently contested improvements observed on calibration using mixup *after* temperature scaling and proposed another improvement of mixup, *MIT*, by generating two sets of mixed samples and then deriving their correct label. We make the same observation of degraded calibration in our study, but propose a different and more efficient approach to preserve it while reaching better performance.

**In regression** The problem of calibration in deep learning has also been studied for regression

tasks (Kuleshov et al., 2018; Song et al., 2019; Laves et al., 2020; Levi et al., 2022), where it is more complex as we lack a simple measure of prediction confidence. In this case, regression models are usually evaluated under the variational inference framework with Monte Carlo (MC) Dropout (Gal & Ghahramani, 2016) to quantify confidence.

In our work, in order to better control the trade-off between adding diversity and uncertainty with mixup, we propose to tailor *interpolation coefficients* to the training data. To do so, we use *warping functions* parameterized by a *similarity kernel* between the points to mix. This allows to mix more strongly similar data and avoid mixing less similar ones, leading to preserving label quality and confidence of the network. We present in Appendix B an empirical analysis on the effect of distance on calibration when mixing, which back up our main assumption that controlling the interpolation through the distance can improve calibration. To keep it efficient, we apply an *online linear interpolation* and mix data from the same batch. As opposed to all other methods discussed above, we also show that our approach is effective both for classification and regression tasks. We present it in detail and the *kernel warping functions* used in the next section.

## 3 KERNEL WARPING MIXUP

### 3.1 PRELIMINARY NOTATIONS AND BACKGROUND

First, we define the notations and elaborate on the learning conditions that will be considered throughout the paper. Let $\mathcal{D} = \{(\mathbf{x}_i, y_i)\}_{i=1}^N = (\mathbf{X}, \mathbf{y}) \in \mathbb{X}^N \times \mathbb{Y}^N \subset \mathbb{R}^{d \times N} \times \mathbb{R}^N$ be the training dataset. We want to learn a *model* $f_\theta$ parameterized by $\theta \in \Theta \subset \mathbb{R}^p$, that predicts $\hat{y} := f_\theta(\mathbf{x})$ for any $\mathbf{x} \in \mathbb{X}$. For classification tasks, we have $\mathbb{Y} \subset \mathbb{R}^c$, and we further assume that the model $f_\theta$ can be separated into an encoder part $h_\varphi$ and classification weights $\mathbf{w} \in \mathbb{R}^c$, such that $\forall \mathbf{x} \in \mathbb{X}, f_\theta(\mathbf{x}) = \mathbf{w}^\top h_\varphi(\mathbf{x})$. To learn our model, we optimize the *weights* of the model $\theta$ in a stochastic manner, by repeating the minimization process of the empirical risk computed on *batch* of data $\mathcal{B}_t = \{(\mathbf{x}_i, y_i)\}_{i=1}^n$ sampled from the training set, for $t \in \{1, \ldots, T\}$ iterations.

With *mixup* (Zhang et al., 2018), at each iteration $t$, the empirical risk is computed on *augmented batch of data* $\tilde{\mathcal{B}}_t = \{(\tilde{\mathbf{x}}_i, \tilde{y}_i)\}_{i=1}^n$, such that $\tilde{\mathbf{x}}_i := \lambda_t \mathbf{x}_i + (1 - \lambda_t)\mathbf{x}_{\sigma_t(i)}$ and $\tilde{y}_i := \lambda_t y_i + (1 - \lambda_t)y_{\sigma_t(i)}$, with $\lambda_t \sim \texttt{Beta}(\alpha, \alpha)$ and $\sigma_t \in \mathfrak{S}_n$ a random permutation of $n$ elements sampled uniformly. Thus, each input is mixed with another input randomly selected from *the same batch*, and $\lambda_t$ represents the strength of the interpolation between them. Besides simplicity, mixing elements within the batch significantly reduces *both memory and computation* costs.

In the following part, we introduce a more general extension of this framework using warping functions, that spans different variants of *mixup*, while preserving its efficiency.

### 3.2 WARPED MIXUP

Towards dynamically changing the interpolation depending on the similarity between points, we rely on *warping functions* $\omega_\tau$, to *warp* interpolation coefficients $\lambda_t$ at every iteration $t$ depending on the parameter $\tau$. These functions $\omega_\tau$ are *bijective transformations* from $[0, 1]$ to $[0, 1]$ defined as such:

$$\omega_\tau(\lambda_t) = \texttt{BetaCDF}(\lambda_t; \tau, \tau) \tag{1}$$

$$= \int_0^{\lambda_t} \frac{u^{\tau-1}(1-u)^{\tau-1}}{B(\tau, \tau)} du, \tag{2}$$

where $\texttt{BetaCDF}$ is the cumulative distribution function (CDF) of the Beta distribution, $B(\tau, \tau)$ is a normalization constant and $\tau \in \mathbb{R}_+^*$ is the *warping parameter* that governs the *strength* and *direction* of the warping. Although the Beta CDF has no closed form solution for non-integer values of its parameters $\alpha$ and $\beta$, accurate approximations

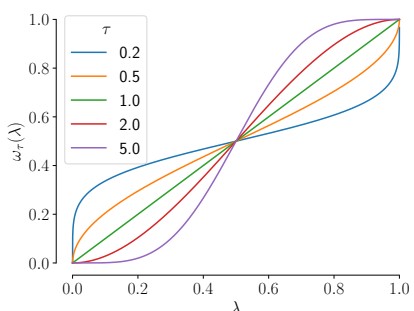

Figure 2: Behavior of $\omega_\tau$ for different values of $\tau$.

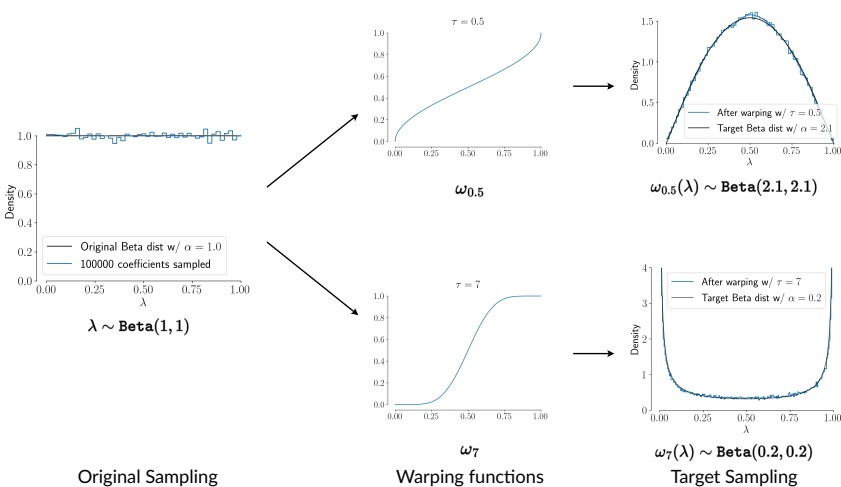

Figure 3: Two examples of coefficients warped to change their underlying distributions. **(Left)** Original sampling of interpolation coefficients $\lambda$ from `Beta(1, 1)`. **(Middle)** Warping functions $\omega_{0.5}$ *(top)* and $\omega_7$ *(bottom)* applied. **(Right)** Resulting distributions of $\omega_{0.5}(\lambda)$ *(top)* and $\omega_7(\lambda)$ *(bottom)*. We can see that $\omega_{0.5}(\lambda)$ closely follows a `Beta(2.1, 2.1)` distribution, and $\omega_7(\lambda)$ a `Beta(0.2, 0.2)`, respectively shown in blue and black lines in the corresponding plots.

are implemented in many statistical software packages.

Our motivation behind such $\omega_\tau$ is to preserve the same type of distribution after warping, *i.e.* Beta distributions with symmetry around 0.5. Similar warping has been used in the *Bayesian Optimization* literature (Snoek et al., 2014), however many other suitable bijection with sigmoidal shape could be considered in our case. Figure 2 illustrates the shape of $\omega_\tau$ and their behavior with respect to $\tau$. These functions have a symmetric behavior around $\tau = 1$ (*in green*), for which warped outputs remain unchanged. When $\tau > 1$ (*in red and purple*) they are pushed towards the extremes (0 and 1), and when $\tau < 1$ (*in orange and blue*), they are pulled towards the center (0.5). We further note that the strength of the warping is *logarithmic* with respect to $\tau$.

Using such warping functions presents the advantage of being able to easily separate the mixing of *inputs* and *targets*, by defining different *warping parameters* $\tau^{(i)}$ and $\tau^{(o)}$. We can now extend the above framework into *warped mixup*:

$$\tilde{\mathbf{x}}_i := \omega_{\tau^{(i)}}(\lambda_t)\mathbf{x}_i + (1 - \omega_{\tau^{(i)}}(\lambda_t))\mathbf{x}_{\sigma_t(i)} \qquad (3)$$

$$\tilde{y}_i := \omega_{\tau^{(o)}}(\lambda_t)y_i + (1 - \omega_{\tau^{(o)}}(\lambda_t))y_{\sigma_t(i)}. \qquad (4)$$

Disentangling *inputs* and *targets* can be interesting when working in the imbalanced setting (Chou et al., 2020). Notably, with $\tau^{(i)} = 1, \tau^{(o)} \approx +\infty$, we recover the Mixup Input Only (IO) variant (Wang et al., 2023) where only inputs are mixed, and with $\tau^{(i)} \approx +\infty, \tau^{(o)} = 1$, the Mixup Target Only (TO) variant (Wang et al., 2023), where only labels are mixed.

Figure 3 presents two examples of warping interpolation coefficients $\lambda$, using two different warping parameters $\tau$ to illustrate the corresponding changes in the underlying distribution of these coefficients. In the following part, we detail our method to select the right $\tau$ depending on the data to mix.

### 3.3 SIMILARITY-BASED KERNEL WARPING

Recall that our goal is to apply stronger interpolation between similar points, and reduce interpolation otherwise, using the warping functions $\omega_\tau$ defined above. Therefore, the parameter $\tau$ should be *exponentially correlated with the distance*, with a symmetric behavior around 1. To this end, we define a class of *similarity kernels*, based on an *inversed, normalized and centered Gaussian kernel*, that outputs the correct warping parameter for the given pair of points. Given a batch of data $\mathbf{x} = \{\mathbf{x}_i\}_{i=1}^n \in \mathbb{R}^{d \times n}$, the index of the first element in the mix $i \in \{1, \ldots, n\}$, along with the

permutation $\sigma \in \mathfrak{S}_n$ to obtain the index of the second element, we compute the following *similarity kernel*:

$$\tau(\mathbf{x}, i, \sigma; \tau_{\max}, \tau_{\text{std}}) = \frac{1}{\tau_{\max}} \exp\left(\frac{\bar{d}_n(\mathbf{x}_i, \mathbf{x}_{\sigma(i)}) - 1}{2\tau_{\text{std}}^2}\right), \tag{5}$$

$$\text{with} \quad \bar{d}_n(\mathbf{x}_i, \mathbf{x}_{\sigma(i)}) = \frac{\sum_{k=1}^d (x_{(k,i)} - x_{(k,\sigma(i))})^2}{\frac{1}{n}\sum_{j=1}^n \sum_{k=1}^d (x_{(k,j)} - x_{(k,\sigma(j))})^2}, \tag{6}$$

where $\bar{d}$ is the squared $L_2$ distance divided by the mean distance over the batch, and $\tau_{\max}, \tau_{\text{std}}$ are respectively the *amplitude* and *standard deviation (std)* of the Gaussian, which are hyperparameters of the similarity kernel. The amplitude $\tau_{\max}$ governs the *strength* of the interpolation *in average*, and $\tau_{\text{std}}$ the *extent* of mixing. Our motivation behind this kernel is to have small values of $\tau$ for small distances and high $\tau$ otherwise, while being able to shut down the mixing effect for points that are too far apart. Figure 4 illustrates the evolution of the density of *warped* interpolation coefficients $\omega_\tau(\lambda)$, depending on the distance between the points to mix. Using this similarity kernel to find the correct $\tau$ to parameterize the warping functions $\omega_\tau$ defines our full *Kernel Warping Mixup* framework. A detailed algorithm of the training procedure can be found in Appendix F.

Note that this exact form of similarity kernel is defined for the warping functions $\omega_\tau$ discussed above and used in the experiments in the next section. Other warping functions might require different kernels depending on their behavior with respect to $\tau$. Likewise, we could consider other similarity measures instead of the squared $L_2$, such as a cosine similarity or an optimal transport metric.

## 4 EXPERIMENTS

We focus our experiments on two very different sets of tasks, namely Image Classification and Regression on Time Series and tabular data. A presentation of the different calibration metrics used can be found in Appendix A. **Image Classification** We mainly follow experimental settings from previous works (Pinto et al., 2022; Wang et al., 2023) and evaluate our approach on CIFAR-10 (C10) and CIFAR-100 (C100) datasets (Krizhevsky et al., 2009) using Resnet34 and Resnet50 architectures (He et al., 2016), and on Tiny-Imagenet (Tiny-IN) (Deng et al., 2009) with a Resnet50. For all our experiments on C10 and C100, we use SGD as the optimizer with a momentum of 0.9 and weight decay of $10^{-4}$, a batch size of 128, and the standard augmentations `random crop`, `horizontal flip` and `normalization`. Models are trained for 200 epochs, with an initial learning rate of 0.1 divided by a factor 10 after 100 and 150 epochs. On Tiny-IN, models are trained for 100 epochs using SGD with an initial learning rate of 0.1 divided by a factor 10 after 40 and 60 epochs, a momentum of 0.9 and weight decay of $10^{-4}$, following (Liu et al., 2022a). We use a batch size of 64 and the same standard augmentations as C10 and C100. We evaluate

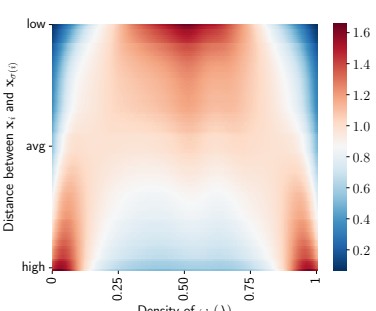

Figure 4: Density of interpolation coefficients *after warping with the similarity kernel $\omega_\tau(\lambda)$ depending on the distance between pairs $(\mathbf{x}_i, \mathbf{x}_{\sigma(i)})$ to mix. Close distances (top lines of the heatmap)* induce strong interpolations, while far distances *(bottom lines of the heatmap)* reduce interpolation.

calibration using ECE (Naeini et al., 2015; Guo et al., 2017), negative log likelihood (NLL) (Hastie et al., 2009) and Brier score (Brier, 1950), after finding the optimal temperature through Temperature Scaling (Guo et al., 2017). Results are reproduced and averaged over 4 different random runs, and we report standard deviation between the runs. For each run, we additionally average the results of the last 10 epochs following (Wang et al., 2023).

**Regression** Here again, we mainly follow settings of previous work on regression (Yao et al., 2022a). We evaluate performance on Airfoil (Kooperberg, 1997), Exchange-Rate and Electricity (Lai et al., 2018) datasets using Root Mean Square Error (RMSE) and Mean Averaged Percentage Error (MAPE), along with Uncertainty Calibration Error (UCE) (Laves et al., 2020) and Expected

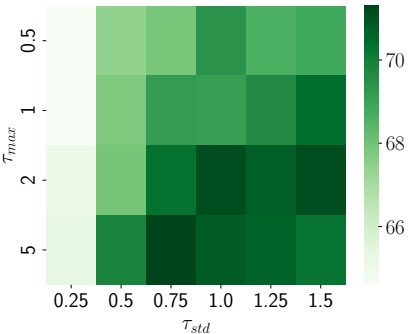 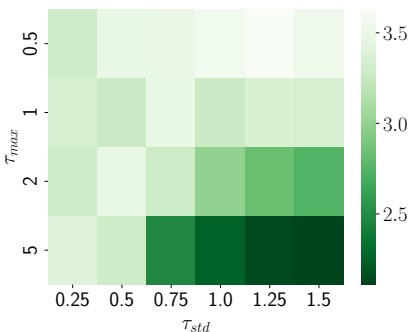

Figure 5: Heatmaps of Accuracy (*left*) and ECE (*right*) from cross validation on C100 for Resnet50.

Table 1: Comparative study on similarity used for Resnet50 on CIFAR10.

| Input similarity | Output similarity | $((\tau_{\max}^{(i)}, \tau_{\text{std}}^{(i)}))$ | $((\tau_{\max}^{(o)}, \tau_{\text{std}}^{(o)}))$ | Accuracy ($\uparrow$) | ECE ($\downarrow$) | Brier ($\downarrow$) | NLL ($\downarrow$) |
|---|---|---|---|---|---|---|---|
| Inputs | Inputs | (2,1) | (2,1) | $95.73 \pm 0.07$ | $0.77 \pm 0.07$ | $6.91 \pm 0.16$ | $17.06 \pm 0.47$ |
| Embedding | Embedding | (2,1) | (2,1) | $95.82 \pm 0.04$ | $0.65 \pm 0.09$ | $6.86 \pm 0.05$ | $17.05 \pm 0.3$ |
| Classif. weights | Classif. weights | (5,0.75) | (5,0.75) | $95.68 \pm 0.07$ | $0.84 \pm 0.1$ | $6.96 \pm 0.11$ | $17.29 \pm 0.31$ |
| Embedding | Classif. weights | (2,1) | (2,1.25) | $95.79 \pm 0.18$ | $0.66 \pm 0.03$ | $6.89 \pm 0.23$ | $17.29 \pm 0.31$ |
| Inputs | Embedding | (2,1) | (2,1) | $95.68 \pm 0.12$ | $0.88 \pm 0.11$ | $6.97 \pm 0.17$ | $16.95 \pm 0.39$ |

Normalized Calibration Error (ENCE) (Levi et al., 2022) for calibration. We train a three-layer fully connected network augmented with Dropout (Srivastava et al., 2014) on Airfoil, and LST-Attn (Lai et al., 2018) on Exchange-Rate and Electricity. All models are trained for 100 epochs with the Adam optimizer (Kingma & Ba, 2014), with a batch size of 16 and learning rate of $0.01$ on Airfoil, and a batch size of 128 and learning rate of $0.001$ on Exchange-Rate and Electricity. To estimate variance for calibration, we rely on MC Dropout (Gal & Ghahramani, 2016) with a dropout of $0.2$ and $50$ samples. Results are reproduced and averaged over 5 different random runs for Exchange Rate and Electricity, and over 10 runs for Airfoil. We also report standard deviation between the runs.

## 4.1 CLASSIFICATION

To find the optimal pair of parameters $(\tau_{\max}, \tau_{\text{std}})$, we conducted cross-validation separately on C10 and C100 datasets. We detail the process of cross-validation in Appendix E. We provide heatmaps of the experiments in Figure 5 for C100, and refer to Appendix E for C10. We can clearly see frontiers and regions of the search spaces that are more optimal than others. In particular, high amplitude and std increase accuracy for both datasets, showing the importance of strong interpolation and not being restrictive in the points to mix. However, while calibration is best when $\tau_{\text{std}}$ is low for C10, good calibration requires that $\tau_{\max}$ and $\tau_{\text{std}}$ are both high for C100. This might reflect the difference in terms of number of class and their separability between both datasets, but a deeper study of the behavior of the confidence would be required.

The flexibility in the framework presented allows to measure similarity between points in any space that can represent them, and also disentangle the similarity used for input and targets. For our experiments in classification, we considered different possible choice:
**(1) Input distance:** we compute the distance between raw input data, *i.e.* $\bar{d}_n(\mathbf{x}_i, \mathbf{x}_{\sigma(i)})$;
**(2) Embedding distance:** we compute the distance between *embeddings* of the input data obtained by the encoder at the current training step, *i.e.* $\bar{d}_n(h_\varphi(\mathbf{x}_i), h_\varphi(\mathbf{x}_{\sigma(i)}))$;
**(3) Classification distance:** we compute the distance between the *classification weights* at the current training step of the class corresponding to the input data, *i.e.* $\bar{d}_n(\mathbf{w}_{y_i}, \mathbf{w}_{y_{\sigma(i)}})$;
In Table 1, we compared results for each of these choice and for different combinations of similarity between inputs and targets. We conducted cross-validation in each case to find the best pairs of parameters $(\tau_{\max}^{(i)}, \tau_{\text{std}}^{(i)})$ for input similarity and $(\tau_{\max}^{(o)}, \tau_{\text{std}}^{(o)})$ for output similarity. We found that results are robust to the choice of similarity considered as difference in performance and calibration are small between them, but using *embedding distance* for both inputs and targets seems to yield the best results. This is the setting chosen for the remaining experiments in classification.

Table 2: Performance (Accuracy in %) and calibration (ECE, Brier, NLL) comparison with Resnet50. Best in **bold**, second best underlined.

| Dataset | Methods | $\alpha$ | Accuracy (↑) | ECE (↓) | Brier (↓) | NLL (↓) |
|---|---|---|---|---|---|---|
| | ERM Baseline | – | 94.26 ± 0.12 | 0.56 ± 0.05 | 8.56 ± 0.23 | 17.93 ± 0.36 |
| | Mixup | 1 | 95.6 ± 0.17 | 1.40 ± 0.12 | 7.13 ± 0.31 | 17.32 ± 0.88 |
| | | 0.5 | 95.53 ± 0.18 | 1.29 ± 0.15 | 7.22 ± 0.30 | 17.44 ± 0.66 |
| | | 0.1 | 94.98 ± 0.25 | 1.29 ± 0.21 | 7.83 ± 0.37 | 17.84 ± 0.78 |
| | Mixup IO | 1 | 94.74 ± 0.34 | **0.47 ± 0.07** | 7.78 ± 0.41 | 16.13 ± 0.75 |
| C10 | | 0.5 | 95.07 ± 0.17 | 0.48 ± 0.08 | 7.39 ± 0.14 | 15.23 ± 0.34 |
| | | 0.1 | 94.79 ± 0.06 | 0.7 ± 0.16 | 7.85 ± 0.20 | 16.37 ± 0.61 |
| | Manifold Mixup | 1 | 96.02 ± 0.08 | 1.32 ± 0.35 | 6.64 ± 0.15 | 16.72 ± 0.34 |
| | | 0.5 | 95.64 ± 0.31 | 1.36 ± 0.11 | 7.15 ± 0.51 | 17.63 ± 1.11 |
| | | 0.1 | 94.79 ± 0.34 | 1.19 ± 0.16 | 8.4 ± 0.54 | 19.77 ± 1.45 |
| | RegMixup | 20 | **96.14 ± 0.15** | 0.91 ± 0.06 | **6.41 ± 0.23** | 14.77 ± 0.33 |
| | MIT-A ($\Delta\lambda > 0.5$) | 1 | 95.68 ± 0.28 | 0.88 ± 0.19 | 6.58 ± 0.43 | **13.88 ± 0.83** |
| | MIT-L ($\Delta\lambda > 0.5$) | 1 | 95.42 ± 0.14 | 0.66 ± 0.08 | 6.85 ± 0.18 | 14.41 ± 0.32 |
| | Kernel Warping Mixup (Ours) | 1 | 95.82 ± 0.04 | 0.65 ± 0.09 | 6.86 ± 0.05 | 17.05 ± 0.3 |
| | ERM Baseline | – | 73.83 ± 0.82 | 2.20 ± 0.13 | 35.90 ± 1.04 | 96.39 ± 3.45 |
| | Mixup | 1 | 78.05 ± 0.23 | 2.41 ± 0.23 | 31.26 ± 0.26 | 88.01 ± 0.53 |
| | | 0.5 | 78.51 ± 0.37 | 2.55 ± 0.22 | 30.44 ± 0.44 | 85.57 ± 1.88 |
| | | 0.1 | 76.49 ± 0.86 | 2.69 ± 0.13 | 32.75 ± 1.05 | 89.82 ± 3.87 |
| | Mixup IO | 1 | 75.25 ± 0.72 | **1.77 ± 0.13** | 34.24 ± 0.68 | 91.41 ± 2.18 |
| C100 | | 0.5 | 76.42 ± 0.81 | 1.94 ± 0.15 | 32.65 ± 1.01 | 86.1 ± 3.04 |
| | | 0.1 | 75.82 ± 0.98 | 2.1 ± 0.22 | 33.45 ± 1.26 | 89.54 ± 3.75 |
| | Manifold Mixup | 1 | **80.39 ± 0.31** | 2.58 ± 0.07 | **28.54 ± 0.34** | **79.06 ± 0.96** |
| | | 0.5 | 79.46 ± 0.91 | 2.76 ± 0.30 | 29.63 ± 1.09 | 82.92 ± 3.43 |
| | | 0.1 | 76.85 ± 1.28 | 2.87 ± 0.28 | 32.54 ± 1.49 | 90.09 ± 4.92 |
| | RegMixup | 10 | 78.44 ± 0.24 | 2.20 ± 0.23 | 30.82 ± 0.29 | 83.16 ± 1.19 |
| | MIT-A ($\Delta\lambda > 0.5$) | 1 | 77.81 ± 0.42 | 2.19 ± 0.05 | 30.84 ± 0.53 | 80.49 ± 1.45 |
| | MIT-L ($\Delta\lambda > 0.5$) | 1 | 77.14 ± 0.71 | 2.13 ± 0.17 | 31.74 ± 1.11 | 82.87 ± 3.24 |
| | Kernel Warping Mixup (Ours) | 1 | 79.62 ± 0.68 | 1.84 ± 0.22 | 29.18 ± 0.78 | 80.46 ± 2.08 |
| | ERM Baseline | - | 66.74 ± 0.34 | 1.62 ± 0.22 | 44.36 ± 0.44 | 135.44 ± 1.94 |
| | Mixup | 1 | 67.21 ± 0.21 | 1.63 ± 0.10 | 44.42 ± 0.29 | 136.67 ± 1.03 |
| | | 0.5 | 67.34 ± 0.69 | 1.56 ± 0.05 | 44.08 ± 1.0 | 135.83 ± 4.36 |
| | | 0.1 | 66.48 ± 0.57 | 1.66 ± 0.17 | 45.32 ± 0.62 | 141.8 ± 2.49 |
| | Mixup IO | 1 | 66.17 ± 0.28 | 1.49 ± 0.21 | 45.02 ± 0.31 | 136.85 ± 0.98 |
| TinyIN | | 0.5 | 66.98 ± 0.39 | 1.75 ± 0.12 | 44.17 ± 0.26 | 134.55 ± 1.3 |
| | | 0.1 | 65.87 ± 0.57 | 1.51 ± 0.21 | 45.5 ± 0.57 | 139.22 ± 2.46 |
| | Manifold Mixup | 1 | 69.49 ± 0.31 | 1.39 ± 0.2 | 41.64 ± 0.33 | 128.85 ± 0.82 |
| | | 0.5 | 68.46 ± 0.24 | 1.57 ± 0.15 | 42.73 ± 0.36 | 132.76 ± 1.38 |
| | | 0.1 | 67.97 ± 0.45 | 1.87 ± 0.08 | 43.37 ± 0.51 | 135.69 ± 1.81 |
| | RegMixup | 20 | **69.71 ± 0.42** | **1.17 ± 0.19** | **40.97 ± 0.68** | **124.31 ± 2.45** |
| | | 10 | 69.39 ± 0.62 | 1.31 ± 0.08 | 41.6 ± 0.77 | 126.77 ± 2.73 |
| | MIT-A ($\Delta\lambda > 0.5$) | 1 | 67.94 ± 0.59 | 1.62 ± 0.26 | 42.89 ± 0.70 | 131.20 ± 1.76 |
| | MIT-L ($\Delta\lambda > 0.5$) | 1 | 67.30 ± 0.96 | 1.7 ± 0.16 | 43.75 ± 1.15 | 134.36 ± 4.23 |
| | Kernel Warping Mixup (Ours) | 1 | 68.18 ± 0.26 | 1.29 ± 0.35 | 43.21 ± 0.47 | 133.01 ± 1.61 |

Then, we present an extensive comparison of results on C10, C100 and Tiny-IN for Resnet50 in Table 2. Results with Resnet34 can be found in Appendix C. We compare our Kernel Warping Mixup with Mixup (Zhang et al., 2018), its variants Mixup-IO (Wang et al., 2023) and Manifold Mixup (Verma et al., 2019), and with the recent RegMixup (Pinto et al., 2022) and MIT (Wang et al., 2023). We can see that our method outperforms in accuracy both Mixup and Mixup IO variants, with better calibration scores in general. It also yields competitive accuracy and calibration with state-of-the-art approaches RegMixup, MIT and Manifold Mixup. In particular, for C100 with Resnet50, it obtains about 1 percentage point (p.p.) higher in accuracy than Mixup, 3 p.p. higher than Mixup-IO, 1.2 p.p. higher than RegMixup and 2 p.p. higher than MIT, while having strong calibration scores. Exact values of $(\tau_{\max}, \tau_{\text{std}})$ used to derive these results are presented in Appendix E.

Table 3: Performance (RMSE, MAPE) and calibration (UCE, ENCE) comparison on several regression tasks. Best in **bold**, second best underlined.

| Dataset | Methods | $\alpha$ | RMSE ($\downarrow$) | MAPE ($\downarrow$) | UCE ($\downarrow$) | ENCE ($\downarrow$) |
|---|---|---|---|---|---|---|
| Airfoil | ERM Baseline | – | 2.843 ± 0.311 | 1.720 ± 0.219 | **107.6 ± 19.179** | 0.0210 ± 0.0078 |
| | Mixup | 0.5 | 3.311 ± 0.207 | 2.003 ± 0.126 | 147.1 ± 33.979 | 0.0212 ± 0.0063 |
| | Manifold Mixup | 0.5 | 3.230 ± 0.177 | 1.964 ± 0.111 | 126.0 ± 15.759 | 0.0206 ± 0.0064 |
| | C-Mixup | 0.5 | 2.850 ± 0.13 | 1.706 ± 0.104 | 111.235 ± 32.567 | 0.0190 ± 0.0075 |
| | Kernel Warping Mixup (Ours) | 0.5 | **2.807 ± 0.261** | **1.694 ± 0.176** | 126.0 ± 23.320 | **0.0180 ± 0.0047** |
| Exch. Rate | ERM Baseline | – | 0.019 ± 0.0024 | 1.924 ± 0.287 | 0.0082 ± 0.0028 | 0.0364 ± 0.0074 |
| | Mixup | 1.5 | 0.0192 ± 0.0025 | 1.926 ± 0.284 | 0.0074 ± 0.0022 | 0.0352 ± 0.0059 |
| | Manifold Mixup | 1.5 | 0.0196 ± 0.0026 | 2.006 ± 0.346 | 0.0086 ± 0.0029 | 0.0382 ± 0.0085 |
| | C-Mixup | 1.5 | 0.0188 ± 0.0017 | 1.893 ± 0.222 | 0.0078 ± 0.0020 | 0.0360 ± 0.0064 |
| | Kernel Warping Mixup (Ours) | 1.5 | **0.0186 ± 0.0020** | **1.872 ± 0.235** | **0.0074 ± 0.0019** | **0.0346 ± 0.0050** |
| Electricity | ERM Baseline | – | 0.069 ± 0.003 | 15.372 ± 0.474 | 0.007 ± 0.001 | **0.219 ± 0.020** |
| | Mixup | 2 | 0.071 ± 0.001 | 14.978 ± 0.402 | **0.006 ± 0.0004** | 0.234 ± 0.012 |
| | Manifold Mixup | 2 | 0.070 ± 0.001 | 14.952 ± 0.475 | 0.007 ± 0.0007 | 0.255 ± 0.015 |
| | C-Mixup | 2 | 0.068 ± 0.001 | **14.716 ± 0.066** | 0.007 ± 0.0006 | 0.233 ± 0.015 |
| | Kernel Warping Mixup (Ours) | 2 | **0.068 ± 0.0006** | 14.827 ± 0.293 | 0.007 ± 0.001 | 0.230 ± 0.013 |

However, our method achieves its competitive results while being much more efficient than the other state-of-the-art approaches. Indeed, our Kernel Warping Mixup is about as fast as Mixup when using input or classification distance, and about $1.5\times$ slower with embedding distance as we have additional computations to obtain the embeddings. However, both RegMixup and MIT are about $2\times$ slower, along with significant memory constraints, since they require training on twice the amount of data per batch which limits the maximum batch size possible in practice. Exact running time comparison can be seen in Appendix D.

## 4.2 REGRESSION

To demonstrate the flexibility of our framework regarding different tasks, we provide experiments on regression for tabular data and time series. Regression tasks have the advantage of having an obvious meaningful distance between points, which is the *label distance*. Since we are predicting continuous values, we can directly measure the similarity between two points by the distance between their labels, *i.e.* $\bar{d}_n(y_i, y_{\sigma(i)})$. This avoids the costly computation of embeddings.

In Table 3, we compare our Kernel Warping Mixup with Mixup (Zhang et al., 2018), Manifold Mixup (Verma et al., 2019) and C-Mixup (Yao et al., 2022a). We can see that our approach achieves competitive results with state-of-the-art C-Mixup, in both performance and calibration metrics. Exact values of $(\tau_{\max}, \tau_{\text{std}})$ used to derive these results are presented in Appendix E. Notably, unlike C-Mixup, our approach do not rely on *sampling rates* calculated before training, which add a lot of computational overhead and are difficult to obtain for large datasets. Furthermore, since we only use elements from within the same batch of data, we also reduce memory usage.

## 5 CONCLUSION

In this paper, we present *Kernel Warping Mixup*, a flexible framework for linearly interpolating data during training, based on warping functions parameterized by a similarity kernel. The coefficients governing the interpolation are warped to change their underlying distribution depending on the similarity between the points to mix. This provides an efficient and strong data augmentation approach that can be applied to different tasks by changing the similarity function depending on the application. We show through extensive experiments the effectiveness of the approach to improve both performance and calibration in classification as well as in regression. It is also worth noting that the proposed framework can be extended by combining it with other Mixup variants such as CutMix (Yun et al., 2019), RegMixup (Pinto et al., 2022) or Manifold Mixup (Verma et al., 2019). Future works include applications to more complex tasks such as semantic segmentation or monocular depth estimation.

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

# A INTRODUCTION TO CALIBRATION METRICS

As discussed in Section 2.2, calibration measures the difference between predictive confidence and actual probability. More formally, with $\hat{y}$ and $y \in \mathbb{Y}$, respectively the model's prediction and target label, and $\hat{p}$ its predicted confidence, a perfectly calibrated model should satisfy $P(\hat{y} = y | \hat{p} = p) = p$, for $p \in [0, 1]$.

We use several metrics for calibration in the paper, namely, ECE, Brier score and NLL for classification tasks, and UCE and ENCE for regression tasks. We formally introduce all of them here.

**NLL** The *negative log-likelihood* (NLL) is a common metric for a model's prediction quality (Hastie et al., 2009). It is equivalent to cross-entropy in multi-class classification. NLL is defined as:

$$\text{NLL}(\mathbf{x}, \mathbf{y}) = -\frac{1}{N} \sum_{i=1}^{N} \log(\hat{p}(\mathbf{y}_i | \mathbf{x}_i)), \tag{7}$$

where $\hat{p}(\mathbf{y}_i | \mathbf{x}_i)$ represents the confidence of the model in the output associated to $\mathbf{x}_i$ for the target class $\mathbf{y}_i$.

**Brier score** The Brier score (Brier, 1950) for multi-class classification is defined as

$$\text{Brier}(\mathbf{x}, \mathbf{y}) = -\frac{1}{N} \sum_{i=1}^{N} \sum_{j=1}^{c} (\hat{p}(y_{(i,j)} | \mathbf{x}_i) - y_{(i,j)})^2, \tag{8}$$

where we assume that the target label $\mathbf{y}_i$ is represented as a one-hot vector over the $c$ possible class, *i.e.*, $\mathbf{y}_i \in \mathbb{R}^c$. Brier score is the mean square error (MSE) between predicted confidence and target.

**ECE** Expected Calibration Error (ECE) is a popular metric for calibration performance for classification tasks in practice. It approximates the difference between accuracy and confidence in expectation by first grouping all the samples into $M$ equally spaced bins $\{B_m\}_{m=1}^{M}$ with respect to their confidence scores, then taking a weighted average of the difference between accuracy and confidence for each bin. Formally, ECE is defined as (Guo et al., 2017):

$$\text{ECE} := \sum_{m=1}^{M} \frac{|B_m|}{N} |\text{acc}(B_m) - \text{conf}(B_m)|, \tag{9}$$

with $\text{acc}(B_m) = \frac{1}{|B_m|} \sum_{i \in B_m} \mathbb{1}_{\hat{y}_i = y_i}$ the accuracy of bin $B_m$, and $\text{conf}(B_m) = \frac{1}{|B_m|} \sum_{i \in B_m} \hat{p}(\mathbf{y}_i | \mathbf{x}_i)$ the average confidence within bin $B_m$.

A probabilistic regression model takes $\mathbf{x} \in \mathbb{X}$ as input and outputs a mean $\mu_y(\mathbf{x})$ and a variance $\sigma_y^2(\mathbf{x})$ targeting the ground-truth $y \in \mathbb{Y}$. The UCE and ENCE calibration metrics are both extension of ECE for regression tasks to evaluate *variance calibration*. They both apply a binning scheme with $M$ bins over the predicted variance.

**UCE** Uncertainty Calibration Error (UCE) (Laves et al., 2020) measures the average of the absolute difference between *mean squared error (MSE)* and *mean variance (MV)* within each bin. It is formally defined by

$$\text{UCE} := \sum_{m=1}^{M} \frac{|B_m|}{N} |\text{MSE}(B_m) - \text{MV}(B_m)|, \tag{10}$$

with $\text{MSE}(B_m) = \frac{1}{|B_m|} \sum_{i \in B_m} (\mu_{y_i}(\mathbf{x}_i) - y_i)^2$ and $\text{MV}(B_m) = \frac{1}{|B_m|} \sum_{i \in B_m} \sigma_{y_i}^2(\mathbf{x}_i)^2$.

Table 4: Performance (Accuracy in %) and calibration (ECE, Brier, NLL) comparison with Resnet34 when mixing only elements higher or lower than a quantile $q$. Best in **bold**, second best underlined.

| Dataset | Quantile of Distance | Accuracy ($\uparrow$) | ECE ($\downarrow$) | Brier ($\downarrow$) | NLL ($\downarrow$) |
|---|---|---|---|---|---|
| | Lower 0.0 / Higher 1.0 (ERM Baseline) | $94.69 \pm 0.27$ | $\mathbf{0.82 \pm 0.11}$ | $8.07 \pm 0.31$ | $17.50 \pm 0.61$ |
| | Lower 1.0 / Higher 0.0 (Mixup) | $95.97 \pm 0.27$ | $1.36 \pm 0.13$ | $6.53 \pm 0.36$ | $16.35 \pm 0.72$ |
| | Lower 0.1 | $95.59 \pm 0.42$ | $\underline{0.88 \pm 0.26}$ | $7.20 \pm 0.56$ | $16.56 \pm 1.52$ |
| | Lower 0.25 | $95.73 \pm 0.18$ | $1.74 \pm 0.45$ | $7.07 \pm 0.26$ | $19.39 \pm 1.11$ |
| C10 | Lower 0.5 | $95.88 \pm 0.28$ | $1.56 \pm 0.28$ | $6.68 \pm 0.34$ | $15.86 \pm 0.71$ |
| | Lower 0.75 | $96.16 \pm 0.09$ | $1.12 \pm 0.16$ | $6.35 \pm 0.15$ | $15.20 \pm 0.44$ |
| | Lower 0.9 | $\mathbf{96.31 \pm 0.08}$ | $1.10 \pm 0.05$ | $\mathbf{6.14 \pm 0.11}$ | $\mathbf{15.16 \pm 0.29}$ |
| | Higher 0.9 | $95.58 \pm 0.34$ | $1.86 \pm 0.25$ | $7.4 \pm 0.48$ | $20.32 \pm 1.25$ |
| | Higher 0.75 | $95.91 \pm 0.14$ | $1.85 \pm 0.17$ | $6.84 \pm 0.22$ | $20.06 \pm 1.12$ |
| | Higher 0.5 | $95.58 \pm 0.28$ | $1.67 \pm 0.13$ | $7.23 \pm 0.37$ | $19.12 \pm 0.74$ |
| | Higher 0.25 | $95.98 \pm 0.3$ | $1.24 \pm 0.18$ | $6.65 \pm 0.51$ | $17.06 \pm 0.99$ |
| | Higher 0.1 | $\underline{96.28 \pm 0.03}$ | $1.13 \pm 0.11$ | $\mathbf{6.14 \pm 0.04}$ | $\underline{15.24 \pm 0.37}$ |
| | Lower 0.0 / Higher 1.0 (ERM Baseline) | $73.47 \pm 1.59$ | $2.54 \pm 0.15$ | $36.47 \pm 2.05$ | $100.82 \pm 6.93$ |
| | Lower 1.0 / Higher 0.0 (Mixup) | $78.11 \pm 0.57$ | $2.49 \pm 0.19$ | $31.06 \pm 0.69$ | $87.94 \pm 1.98$ |
| | Lower 0.1 | $75.40 \pm 0.53$ | $3.48 \pm 0.24$ | $35.92 \pm 0.5$ | $105.87 \pm 1.41$ |
| | Lower 0.25 | $77.14 \pm 0.51$ | $2.54 \pm 0.22$ | $32.95 \pm 0.62$ | $95.42 \pm 1.78$ |
| C100 | Lower 0.5 | $77.66 \pm 0.15$ | $\mathbf{1.85 \pm 0.43}$ | $31.94 \pm 0.28$ | $89.97 \pm 1.53$ |
| | Lower 0.75 | $78.43 \pm 0.62$ | $1.95 \pm 0.6$ | $30.64 \pm 0.7$ | $85.06 \pm 1.89$ |
| | Lower 0.9 | $\mathbf{79.24 \pm 0.7}$ | $1.99 \pm 0.03$ | $\underline{29.72 \pm 0.94}$ | $\underline{82.54 \pm 2.82}$ |
| | Higher 0.9 | $77.3 \pm 0.43$ | $\underline{1.92 \pm 0.22}$ | $32.0 \pm 0.59$ | $88.69 \pm 1.67$ |
| | Higher 0.75 | $77.8 \pm 1.05$ | $2.29 \pm 0.24$ | $31.48 \pm 1.18$ | $88.16 \pm 3.86$ |
| | Higher 0.5 | $78.74 \pm 0.43$ | $2.52 \pm 0.22$ | $30.37 \pm 0.56$ | $84.64 \pm 1.63$ |
| | Higher 0.25 | $78.51 \pm 84.64$ | $2.34 \pm 0.26$ | $30.42 \pm 0.59$ | $84.64 \pm 2.23$ |
| | Higher 0.1 | $\underline{79.14 \pm 0.53}$ | $2.23 \pm 0.34$ | $\mathbf{29.62 \pm 0.51}$ | $\mathbf{82.22 \pm 1.28}$ |

**ENCE** Expected Normalized Calibration Error (ENCE) (Levi et al., 2022) measures the absolute *normalized* difference, between *root mean squared error (RMSE)* and *root mean variance (RMV)* within each bin. It is formally defined by

$$\text{ENCE} := \frac{1}{M} \sum_{m=1}^{M} \frac{|\text{RMSE}(B_m) - \text{RMV}(B_m)|}{\text{RMV}(B_m)}, \tag{11}$$

with $\text{RMSE}(B_m) = \sqrt{\frac{1}{|B_m|} \sum_{i \in B_m} (\mu_{y_i}(\mathbf{x}_i) - y_i)^2}$ and $\text{RMV}(B_m) = \sqrt{\frac{1}{|B_m|} \sum_{i \in B_m} \sigma_{y_i}^2(\mathbf{x}_i)^2}$.

## B    EFFECT OF DISTANCE ON CALIBRATION

In this section, we provide an empirical study of the effect of distance on calibration. The assumption underlying this work is that there is a trade-off in data augmentation procedures between adding *diversity* and introducing *uncertainty* through *manifold intrusion* (Guo et al., 2019; Baena et al., 2022), *i.e*, conflicts between the synthetic labels of the mixed-up examples and the labels of original training data. To show that, we conduct in Table 4 an empirical analysis that compares accuracy and calibration metrics when mixing only pairs of points with distances lower or higher than a given quantile of the overall pairwise distances within the batch, using a Resnet34 on CIFAR10 and CIFAR100 datasets. Note that one should compare results of "Lower $q$" with "Higher $1 - q$" to have equivalent numbers of possible element to mix with (*diversity*). We can see that, in general, **mixing pairs with lower distances leads to better calibration than mixing pairs with higher distances**. These results confirm that there is a trade-off between adding diversity and uncertainty with data augmentation.

## C    FULL RESULTS WITH RESNET34

We present in Table 5 results with Resnet34 on C10 and C100 datasets. We can see that our proposed approach achieves competitive results both in terms of performance and calibration.

Table 5: Performance (Accuracy in %) and calibration (ECE, Brier, NLL) comparison with Resnet34. Best in **bold**, second best underlined.

| Dataset | Methods | $\alpha$ | Accuracy (↑) | ECE (↓) | Brier (↓) | NLL (↓) |
|---|---|---|---|---|---|---|
| | ERM Baseline | – | $94.69 \pm 0.27$ | $0.82 \pm 0.11$ | $8.07 \pm 0.31$ | $17.50 \pm 0.61$ |
| | Mixup | 1 | $95.97 \pm 0.27$ | $1.36 \pm 0.13$ | $6.53 \pm 0.36$ | $16.35 \pm 0.72$ |
| | | 0.5 | $95.71 \pm 0.26$ | $1.33 \pm 0.08$ | $7.03 \pm 0.46$ | $17.47 \pm 1.18$ |
| | | 0.1 | $95.37 \pm 0.22$ | $1.13 \pm 0.11$ | $7.37 \pm 0.36$ | $17.43 \pm 0.79$ |
| C10 | Mixup IO | 1 | $95.16 \pm 0.22$ | $0.6 \pm 0.11$ | $7.3 \pm 0.33$ | $15.56 \pm 0.67$ |
| | | 0.5 | $95.31 \pm 0.17$ | $0.58 \pm 0.06$ | $7.12 \pm 0.21$ | $15.09 \pm 0.45$ |
| | | 0.1 | $95.12 \pm 0.21$ | $0.7 \pm 0.09$ | $7.38 \pm 0.27$ | $15.76 \pm 0.55$ |
| | RegMixup | 20 | $\mathbf{96.51 \pm 0.2}$ | $0.76 \pm 0.08$ | $\mathbf{5.78 \pm 0.26}$ | $\mathbf{13.14 \pm 0.47}$ |
| | MIT-A ($\Delta\lambda > 0.5$) | 1 | $95.78 \pm 0.22$ | $1.02 \pm 0.19$ | $6.51 \pm 0.29$ | $14.04 \pm 0.67$ |
| | MIT-L ($\Delta\lambda > 0.5$) | 1 | $95.71 \pm 0.06$ | $0.67 \pm 0.12$ | $6.57 \pm 0.12$ | $\underline{13.89 \pm 0.28}$ |
| | Kernel Warping Mixup (Ours) | 1 | $\underline{96.16 \pm 0.09}$ | $\mathbf{0.51 \pm 0.07}$ | $6.39 \pm 0.16$ | $16.59 \pm 0.55$ |
| | ERM Baseline | – | $73.47 \pm 1.59$ | $2.54 \pm 0.15$ | $36.47 \pm 2.05$ | $100.82 \pm 6.93$ |
| | Mixup | 1 | $78.11 \pm 0.57$ | $2.49 \pm 0.19$ | $31.06 \pm 0.69$ | $87.94 \pm 1.98$ |
| | | 0.5 | $77.14 \pm 0.67$ | $2.7 \pm 0.36$ | $32.01 \pm 0.93$ | $91.22 \pm 3.05$ |
| | | 0.1 | $76.01 \pm 0.62$ | $2.54 \pm 0.24$ | $33.41 \pm 0.57$ | $93.96 \pm 1.76$ |
| C100 | Mixup IO | 1 | $74.44 \pm 0.49$ | $2.02 \pm 0.14$ | $35.25 \pm 0.43$ | $96.5 \pm 1.62$ |
| | | 0.5 | $74.45 \pm 0.6$ | $1.94 \pm 0.09$ | $35.2 \pm 0.58$ | $96.75 \pm 1.89$ |
| | | 0.1 | $74.21 \pm 0.46$ | $2.39 \pm 0.11$ | $35.38 \pm 0.48$ | $98.24 \pm 1.81$ |
| | RegMixup | 10 | $\underline{78.49 \pm 0.35}$ | $\mathbf{1.64 \pm 0.14}$ | $30.42 \pm 0.26$ | $\mathbf{82.20 \pm 0.78}$ |
| | MIT-A ($\Delta\lambda > 0.5$) | 1 | $77.39 \pm 0.32$ | $2.38 \pm 0.14$ | $31.37 \pm 0.46$ | $83.08 \pm 1.38$ |
| | MIT-L ($\Delta\lambda > 0.5$) | 1 | $76.51 \pm 0.33$ | $2.54 \pm 0.16$ | $32.62 \pm 0.28$ | $86.81 \pm 0.89$ |
| | Kernel Warping Mixup (Ours) | 1 | $\mathbf{79.13 \pm 0.44}$ | $\underline{1.75 \pm 0.44}$ | $\mathbf{29.59 \pm 0.52}$ | $\underline{82.88 \pm 1.32}$ |

Table 6: Training time comparison (in seconds) for a single epoch of CIFAR10 with a Resnet50, measured on a single NVIDIA A100 GPU.

| Method | Time per epoch (in s) |
|---|---|
| Mixup | 20 |
| Manifold Mixup | 33 |
| RegMixup | 38 |
| MIT-L | 40 |
| MIT-A | 46 |
| Kernel Warping Mixup - Input | 21 |
| Kernel Warping Mixup - Classif. | 22 |
| Kernel Warping Mixup - Embedding | 28 |

## D  EFFICIENCY COMPARISON

Table 6 presents comparison of training time for a single epoch on CIFAR10 with a Resnet50. We can see that our Kernel Warping Mixup is about as fast as Mixup when using input or classification distance, and about $1.5\times$ slower with embedding distance as we have additional computations to obtain the embeddings, while both RegMixup and MIT are about $2\times$ slower.

## E  CROSS VALIDATION

We detail the cross-validation settings that we use here. For all different experimental settings, we search for the best $\tau_{max}$ and $\tau_{std}$ *before training*, by conducting cross-validation with a stratified sampling on a 90/10 split of the training set, similarly to Pinto et al. (2022), and average the results across 4 different splits. The hyperparameters are selected to have a good trade-off between calibration

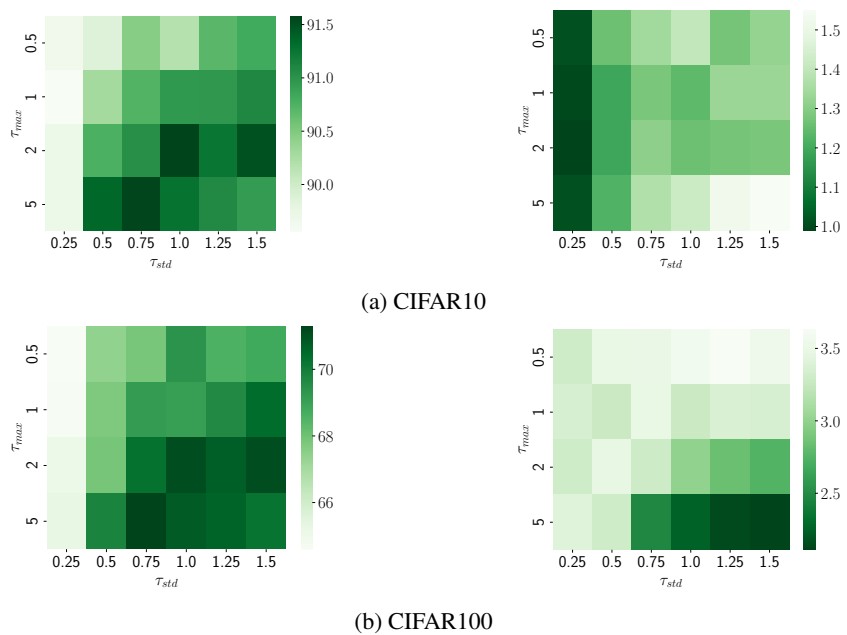

Figure 6: Heatmaps of Accuracy (*left*) and ECE (*right*) from cross validation on CIFAR10 (*top*) and CIFAR100 (*bottom*) for Resnet50.

Table 7: Parameters of the kernel similarity found through cross-validation for each dataset.

| Hyperparameter | Model and Dataset | | | | | | | |
| --- | --- | --- | --- | --- | --- | --- | --- | --- |
| | Resnet34 | | Resnet50 | | | Airfoil | Exchange Rate | Electricity |
| | C10 | C100 | C10 | C100 | Tiny-IN | | | |
| $\tau_{max}$ | 0.5 | 1 | 2 | 2 | 2 | 0.0001 | 500 | 0.001 |
| $\tau_{std}$ | 1.25 | 0.75 | 1 | 1 | 1.25 | 1.5 | 1 | 1.5 |

and accuracy. The optimal values vary between datasets, since they depend on the statistics of the pairwise distances. In Table 7, we present the hyperparameters found through cross-validation for each dataset and model. These values were used to obtain the results discussed in Section 4. We provide separate heatmaps of cross-validation for both CIFAR10 and CIFAR100 datasets with Resnet50 in Figure 6, and repeat our observations here. High amplitude and std increase accuracy for both datasets, showing the importance of strong interpolation and not being restrictive in the points to mix. However, while calibration is best when $\tau_{std}$ is low for C10, good calibration requires that $\tau_{max}$ and $\tau_{std}$ are both high for C100. This might reflect the difference in terms of number of class and their separability between both datasets, but a deeper study of the behavior of the confidence would be required.

## F    DETAILED ALGORITHM

We present a pseudocode of our *Kernel Warping Mixup* procedure for a single training iteration in Algorithm 1. The generation of new data is explained in the pseudocode as a sequential process for simplicity and ease of understanding, but the actual implementation is optimized to work in parallel on GPU through vectorized operations.

## G    ON THE DEPENDENCE TO BATCH SIZE STATISTICS

---

**Algorithm 1:** Kernel Warping Mixup training procedure

---

**Input:** Batch of data $\mathcal{B} = \{(\mathbf{x}_i, y_i)\}_{i=1}^n$, mixup parameter $\alpha$, similarity parameters $(\tau_{\max}, \tau_{\mathrm{std}})$,
    parameters of the model at the current iteration $\theta_t$

$\tilde{\mathcal{B}} \leftarrow \varnothing$
$\sigma_t \sim \mathfrak{S}_n$                                            // Sample random permutation
**for** $\forall i \in \{1, \ldots, n\}$ **do**
    $\lambda_i \sim \mathtt{Beta}(\alpha, \alpha)$
    // Compute warping parameters for inputs and targets
        separately through Equations (5) and (6)
    $\tau_i^{(i)} := \tau^{(i)}(\mathbf{x}, i, \sigma; \tau_{\max}^{(i)}, \tau_{\mathrm{std}}^{(i)})$
    $\tau_i^{(o)} := \tau^{(o)}(\mathbf{x}, i, \sigma; \tau_{\max}^{(o)}, \tau_{\mathrm{std}}^{(o)})$
    // Generate new data
    $\tilde{\mathbf{x}}_i := \omega_{\tau_i^{(i)}}(\lambda_i)\mathbf{x}_i + (1 - \omega_{\tau_i^{(i)}}(\lambda_i))\mathbf{x}_{\sigma(i)}$
    $\tilde{y}_i := \omega_{\tau_i^{(o)}}(\lambda_i)y_i + (1 - \omega_{\tau_i^{(o)}}(\lambda_i))y_{\sigma(i)}$
    // Aggregate new batch
    $\tilde{\mathcal{B}} \leftarrow \tilde{\mathcal{B}} \cup (\tilde{\mathbf{x}}_i, \tilde{y}_i)$

Compute and optimize loss over $\tilde{\mathcal{B}}$
**Output:** updated parameters of the model $\theta_{t+1}$

---

Table 8: Performance (RMSE, MAPE) and calibration (UCE, ENCE) comparison on several regression tasks. Best in **bold**, second best underlined.

| Dataset | Methods | $\alpha$ | RMSE ($\downarrow$) | MAPE ($\downarrow$) | UCE ($\downarrow$) | ENCE ($\downarrow$) |
|---|---|---|---|---|---|---|
| Airfoil | ERM Baseline | – | $2.843 \pm 0.311$ | $1.720 \pm 0.219$ | $\mathbf{107.6 \pm 19.179}$ | $0.0210 \pm 0.0078$ |
| | Mixup | 0.5 | $3.311 \pm 0.207$ | $2.003 \pm 0.126$ | $147.1 \pm 33.979$ | $0.0212 \pm 0.0063$ |
| | Manifold Mixup | 0.5 | $3.230 \pm 0.177$ | $1.964 \pm 0.111$ | $126.0 \pm 15.759$ | $0.0206 \pm 0.0064$ |
| | C-Mixup | 0.5 | $2.850 \pm 0.13$ | $\underline{1.706 \pm 0.104}$ | $111.235 \pm 32.567$ | $\underline{0.0190 \pm 0.0075}$ |
| | Batch - Kernel Warping Mixup (Ours) | 0.5 | $\mathbf{2.807 \pm 0.261}$ | $\mathbf{1.694 \pm 0.176}$ | $126.0 \pm 23.320$ | $\mathbf{0.0180 \pm 0.0047}$ |
| | Global - Kernel Warping Mixup (Ours) | $2.568 \pm 0.235$ | $\mathbf{1.529 \pm 0.143}$ | $95.553 \pm 14.737$ | $\mathbf{0.0155 \pm 0.0046}$ | |
| Exch. Rate | ERM Baseline | – | $0.019 \pm 0.0024$ | $1.924 \pm 0.287$ | $0.0082 \pm 0.0028$ | $0.0364 \pm 0.0074$ |
| | Mixup | 1.5 | $0.0192 \pm 0.0025$ | $1.926 \pm 0.284$ | $\underline{0.0074 \pm 0.0022}$ | $\underline{0.0352 \pm 0.0059}$ |
| | Manifold Mixup | 1.5 | $0.0196 \pm 0.0026$ | $2.006 \pm 0.346$ | $0.0086 \pm 0.0029$ | $0.0382 \pm 0.0085$ |
| | C-Mixup | 1.5 | $\underline{0.0188 \pm 0.0017}$ | $\underline{1.893 \pm 0.222}$ | $0.0078 \pm 0.0020$ | $0.0360 \pm 0.0064$ |
| | Batch - Kernel Warping Mixup (Ours) | 1.5 | $\mathbf{0.0186 \pm 0.0020}$ | $\mathbf{1.872 \pm 0.235}$ | $\mathbf{0.0074 \pm 0.0019}$ | $\mathbf{0.0346 \pm 0.0050}$ |
| | Global - Kernel Warping Mixup (Ours) | 1.5 | $\mathbf{0.0186 \pm 0.0020}$ | $1.875 \pm 0.236$ | $\mathbf{0.0074 \pm 0.0019}$ | $\mathbf{0.0346 \pm 0.0050}$ |

The distance $\bar{d}_n$ presented in Section 3.3 depends on the batch statistics. The rationale behind the normalization by the mean is mainly to rescale the distances to similar scales between tasks and models. If the batch size is too small, one can normalize the distance with the *median* instead of the *mean* to make it more robust to outliers. We did not observe difference in results between the two in our experiments. Then, if one can use a meaningful distance between data points different from the distance in the embedding space of the model to train, statistics such as mean or median of the pairwise similarity matrix can be computed beforehand to normalize the distance during training. In our experiments on regression tasks, we also compared with normalizing using the mean over the whole training set instead of the batch. We found slightly improved results on Airfoil, for which the batch size is 16, but no meaningful improvements for the two other datasets, for which the batch size is 128. These results are provided in Table 8.

