# OpenReview forum: "Tailoring Mixup to Data using Kernel Warping functions"
_ICLR.cc/2024/Conference — Submitted to ICLR 2024_

### Official Review · Reviewer_djAy · 2023-10-24

**Soundness:** 3 good
**Presentation:** 4 excellent
**Contribution:** 2 fair
**Rating:** 6
**Confidence:** 3

**Summary:**

The paper presents a Kernel Warping Mixup technique toward a flexible distribution adjustment by considering the sample similarity. The proposed method changes the underlying distributions used for sampling interpolation coefficients by defining warping functions, allowing inputs and labels to be disentangled, and providing a framework that encompasses several variants of Mixup. To this end, the author introduces a similarity kernel that considers the distance between points when selecting a parameter for the warping function. This paper demonstrates the applicability of the Kernel Warping Mixup across classification and regression tasks, and cites improvements in performance, calibration, and efficiency while it requires less computation, additionally.

**Strengths:**

+ The paper provides a comprehensive analysis and comparison of related work in the field of data augmentation. The organization of the related work into different topics provides all-sided evaluations of the contributions.

+ The proposed idea is somehow novel since few existing works link the coefficients of interpolation with data geometry.

+ The proposed method has been proven effective in improving the model calibration while the improvement of in-distribution generalization is not very significant.

**Weaknesses:**

- Compared to RegMixup, the proposed method brings marginal improvement in classification task accuracies on different models (Tables 2 and 3). Plus, the benchmark result of the Manifold Mixup in classification is missing. The concern is on the expressivity of the proposed method since general data augmentation aims to enrich the diversity of data and cover more uncertain data points as much as possible instead of encouraging similar data to be mixed more.

- The warped Gaussian Kernel into the original Beta distribution for the mixup coefficient $\lambda$ highly depends on batched data similarity. If the batch size is small (for large-scale tasks), the relative distances among the data will collapse, i.e. the same samples will be mixup with different $\lambda$ whether an outlier appears. In Eqn (6), if $n$ is small, the total distance is sensitive to the largest sample distance. It weakens the robustness of the proposed method.

- The evaluation and explanation of how the proposed method enhances calibration over vanilla Mixup are insufficient. For example, (Thulasidasan et al., 2019) have concluded that label smoothing in Mixup training significantly contributes to improved calibration, supported by comprehensive observations such as "score of winning class along the time" and "overconfidence". C-Mixup (Yao et al., 2022a) provides the theoretical justification for Theorem 3. To enhance the comparison, it would be better to establish a more meaningful connection between the proposed method and calibration, moving beyond only the presentation of ECE and Brier scores.

=====================Typos==============

- Missing reference to the initial mention of "MIT" in Section 2.2.
- In Eqn(6), the LHS should indicate the dependence of $n$ if the measure on RHS depends on all $n$ samples.

**Questions:**

1. Since general data augmentation aims to enrich the diversity of data and cover more uncertain data points, is adjusting the distribution of coefficients based on data distribution reasonable to enhance the expressivity? Please see weakness 1.

2. Why choose different benchmarks between Tables 3 and 4, since Manifold Mixup can also applied to classification problems?

3. As suggested by (Wang et al., 2023), Mixup fails to improve the calibration in terms of other metrics such as Calibrated ECE and Optimal ECE. Can the author explain more or provide more evidence that the method in this paper does help the calibration? Why does altering the underlying distribution of $\lambda$ yield a better calibration over vanilla Mixup, even with the same similarity measure of input and output (Row 2 in Table 1)?

I am keenly awaiting the author's response to see how they address my concerns, and I am open to increasing my score based on the response.

**Details Of Ethics Concerns:**

No ethical concerns, since interpolation methods use accessible authorised data.

---

> ### Author Response · Authors · 2023-11-21
> **Response to Reviewer djAy 1/2**
>
> We thank the reviewer for their constructive feedbacks on our work.
>
> - W1:
>   - 1. **Comparison with RegMixup:** We would like to refer the reviewer to Appendix B in which we show that compared to both RegMixup and MIT, our proposed approach is much more efficient both in terms of computation time and memory usage. Our method relies on a single batch of augmented data, while the augmentations in RegMixup and MIT double the size of the batch, leading to more compute and more required memory, as discussed in section 4.1 in the paper.
>   - 2. **Manifold Mixup on classification (Table 1):** We reproduced results of Manifold Mixup (Verma et al., ICML 2019) on CIFAR 10 and CIFAR100 using a Resnet50. We achieve similar performance improvements, but with better calibration. As for RegMixup, we can consider combining our method with Manifold Mixup in future work.
>   - 3. **Data augmentation aims to cover uncertain data points (Table 2):** The assumption underlying this work is that there is a trade-off in data augmentation procedures between adding *diversity* and introducing *uncertainty*. While we agree that it is important to cover more regions of the space, it is important to do it without adding errors and avoid *manifold intrusion* (Guo et al., 2019; Baena et al., 2022), *i.e*, conflicts between the synthetic labels of the mixed-up examples and the labels of original training data. Our intuition is that mixing data that are too far introduces uncertainty on the true underlying label of the new point, for which a linear combination of the labels might not be an accurate target. We confirm this hypothesis by the following experiment that we report in Table 2, in which we compare *accuracy* and *calibration* metrics when mixing *only* pairs of points with distances lower or higher than a given quantile of the overall pairwise distances within the batch, using a Resnet34 on CIFAR10 and CIFAR100 datasets. Note that one should compare results of "Lower $q$" with "Higher $1 - q$" to have equivalent numbers of possible element to mix with (*diversity*). We can see that, in general, *mixing pairs with lower distances leads to better calibration than mixing pairs with higher distances* (*uncertainty*). The introduction of uncertainty when mixing pairs with high distance can be explained by *manifold intrusion* (Guo et al., 2019; Baena et al., 2022). These results confirm that there is a trade-off between adding *diversity* and *uncertainty with* data augmentation.
>
> - W2. This is an interesting remark for which several points need to be discussed.
>   - 1. **Statistics can be computed before training (Table 3):** First, if one can use a meaningful distance between data points different from the distance in the embedding space of the model to train, statistics such as mean or median of the pairwise similarity matrix can be computed beforehand to normalize the distance during training. In our experiments on regression tasks, we also compared with normalizing using the mean over the whole training set instead of the batch. We found slightly improved results on Airfoil, for which the batch size is 16, but no meaningful improvements for the two other datasets, for which the batch size is 128. These results are provided in Table 3.
>   - 2. **Median can be used to be more robust to outliers:** Second, our rationale behind the normalization by the mean is mainly to rescale the distances to similar scales between tasks and models, and same for $\tau_{max}$ and $\tau_{std}$. If the batch size is too small, one could use the median instead of the mean to be more robust to outliers.
>   - 3. **Distance impact distribution of $\lambda$:** Third, regarding the remark "the same samples will be mixup with different $\lambda$ whether an outlier appears", we would like to point that since we are sampling different $\lambda$ at each iteration, it is very likely that the same pair of samples will be mixed with different $\lambda$ at each iteration, whatever the mixup approach we are using. In our case, the *underlying distribution* of the $\lambda$ might vary for the same pair of samples between iterations depending on the distances in each batch, but we do not think that it hinders the robustness of the approach. If the distance between two samples is high, it is *more likely* that the $\alpha$ parameter will be small and thus that $\lambda$ will be close to 0 and 1, even if the batch size is small.
>
> - W3. **See W1.3** We provide in Table 2 an analysis to back up our intuition that mixing data that are too far introduces uncertainty on the true underlying label of the new point, and that a linear combination of the labels might not be an accurate target.

---

> > ### Author Response · Authors · 2023-11-21
> > **Response to Reviewer djAy 2/2**
> >
> > - Q1. **See W1.3**
> > - Q2. **See W1.2:** We followed the settings of C-Mixup for our benchmark on regression tasks, for which Manifold Mixup was easily implemented. We provide results for Manifold Mixup on classification tasks in Table 1.
> > - Q3. **Optimal ECE are compared:** Indeed as mentioned by Wang et al. (2023), but also Ashuka et al. (2020), one should compare ECE after calibration through temperature scaling instead of raw ECE, since the ordering of results can change between the two. That's why we report *Optimal ECE* results after temperature scaling.
> >
> > - Typos:
> >   - The MIT method is introduced in Wang et al. (2023), that we cite in the same sentence. We will make it clearer in the revised version.
> >   - We will add the mention of the dependence on $n$ in the LHS.
> >
> > ### References:
> > - Guo et al., Mixup as locally linear out-of-manifold regularization. AAAI, 2019.
> > - Baena et al., A local mixup to prevent manifold intrusion. EUSIPCO, 2022.
> > - Wang et al. On the Pitfall of Mixup for Uncertainty Calibration. CVPR 2023.
> > - Ashuka et al. Pitfalls of In-domain Uncertainty Estimation and Ensembling in Deep Learning. ICLR 2020.

---

> > > ### Author Response · Authors · 2023-11-21
> > > **Table of Results 1/2**
> > >
> > > ### Table 1: Performance and calibration comparison of Manifold Mixup on C10 and C100 with Resnet50
> > >
> > > | Dataset | Methods | $\alpha$ | Accuracy | ECE | Brier | NLL |
> > > | --- | --- | :---: | :---: | :---: | :---: | :---: |
> > > |  C10 | ERM Baseline | -- | 94.26 $\pm$ 0.12 | 0.56 $\pm$ 0.05 | 8.56 $\pm$ 0.23 | 17.93 $\pm$ 0.36
> > > |  | Mixup | 1 | 95.6 $\pm$ 0.17 | 1.40 $\pm$ 0.12 | 7.13 $\pm$ 0.31 | 17.32 $\pm$ 0.88
> > > |  |  | 0.5 | 95.53 $\pm$ 0.18 | 1.29 $\pm$ 0.15 | 7.22 $\pm$ 0.30 | 17.44 $\pm$ 0.66
> > > |  |  | 0.1 | 94.98 $\pm$ 0.25 | 1.29 $\pm$ 0.21 | 7.83 $\pm$ 0.37 | 17.84 $\pm$ 0.78
> > > |  | Mixup IO | 1 | 94.74 $\pm$ 0.34 | 0.47 $\pm$ 0.07 | 7.78 $\pm$ 0.41 | 16.13 $\pm$ 0.75
> > > |  |  | 0.5 | 95.07 $\pm$ 0.17 | 0.48 $\pm$ 0.08 | 7.39 $\pm$ 0.14 | 15.23 $\pm$ 0.34
> > > |  |  | 0.1 | 94.79 $\pm$ 0.06 | 0.7 $\pm$ 0.16 | 7.85 $\pm$ 0.20 | 16.37 $\pm$ 0.61
> > > |  | Manifold Mixup | 2 | 95.86 $\pm$ 0.18 | 1.04 $\pm$ 0.23 | 6.93 $\pm$ 0.27 | 17.18 $\pm$ 0.66
> > > |  |  | 1 | 96.02 $\pm$ 0.08 | 1.32 $\pm$ 0.35 | 6.64 $\pm$ 0.15 | 16.72 $\pm$ 0.34
> > > |  |  | 0.5 | 95.64 $\pm$ 0.31 | 1.36 $\pm$ 0.11 | 7.15 $\pm$ 0.51 | 17.63 $\pm$ 1.11
> > > |  |  | 0.1 | 94.79 $\pm$ 0.34 | 1.19 $\pm$ 0.16 | 8.4 $\pm$ 0.54 | 19.77 $\pm$ 1.45
> > > |  | RegMixup | 20 | 96.14 $\pm$ 0.15 | 0.91 $\pm$ 0.06 | 6.41 $\pm$ 0.23 | 14.77 $\pm$ 0.33
> > > |  | MIT-A ($\Delta \lambda > 0.5$) | 1 | 95.68 $\pm$ 0.28 | 0.88 $\pm$ 0.19 | 6.58 $\pm$ 0.43 | 13.88 $\pm$ 0.83
> > > |  | MIT-L ($\Delta \lambda > 0.5$) | 1 | 95.42 $\pm$ 0.14 | 0.66 $\pm$ 0.08 | 6.85 $\pm$ 0.18 | 14.41 $\pm$ 0.32
> > > |  | Kernel Warping Mixup (Ours) | 1 | 95.82 $\pm$ 0.04 | 0.65 $\pm$ 0.09 | 6.86 $\pm$ 0.05 | 17.05 $\pm$ 0.3
> > > |  C100 | ERM Baseline | -- | 73.83 $\pm$ 0.82 | 2.20 $\pm$ 0.13 | 35.90 $\pm$ 1.04 | 96.39 $\pm$ 3.45
> > > |  | Mixup | 1 | 78.05 $\pm$ 0.23 | 2.41 $\pm$ 0.23 | 31.26 $\pm$ 0.26 | 88.01 $\pm$ 0.53
> > > |  |  | 0.5 | 78.51 $\pm$ 0.37 | 2.55 $\pm$ 0.22 | 30.44 $\pm$ 0.44 | 85.57 $\pm$ 1.88
> > > |  |  | 0.1 | 76.49 $\pm$ 0.86 | 2.69 $\pm$ 0.13 | 32.75 $\pm$ 1.05 | 89.82 $\pm$ 3.87
> > > |  | Mixup IO | 1 | 75.25 $\pm$ 0.72 | 1.77 $\pm$ 0.13 | 34.24 $\pm$ 0.68 | 91.41 $\pm$ 2.18
> > > |  |  | 0.5 | 76.42 $\pm$ 0.81 | 1.94 $\pm$ 0.15 | 32.65 $\pm$ 1.01 | 86.1 $\pm$ 3.04
> > > |  |  | 0.1 | 75.82 $\pm$ 0.98 | 2.1 $\pm$ 0.22 | 33.45 $\pm$ 1.26 | 89.54 $\pm$ 3.75
> > > |  | Manifold Mixup | 2 | 81.54 $\pm$ 0.12 | 2.2 $\pm$ 0.14 | 26.94 $\pm$ 0.15 | 73.15 $\pm$ 0.75
> > > |  |  | 1 | 80.39 $\pm$ 0.31 | 2.58 $\pm$ 0.07 | 28.54 $\pm$ 0.34 | 79.06 $\pm$ 0.96
> > > |  |  | 0.5 | 79.46 $\pm$ 0.91 | 2.76 $\pm$ 0.30 | 29.63 $\pm$ 1.09 | 82.92 $\pm$ 3.43
> > > |  |  | 0.1 | 76.85 $\pm$ 1.28 | 2.87 $\pm$ 0.28 | 32.54 $\pm$ 1.49 | 90.09 $\pm$ 4.92
> > > |  | RegMixup | 10 | 78.44 $\pm$ 0.24 | 2.20 $\pm$ 0.23 | 30.82 $\pm$ 0.29 | 83.16 $\pm$ 1.19
> > > |  | MIT-A ($\Delta \lambda > 0.5$) | 1 | 77.81 $\pm$ 0.42 | 2.19 $\pm$ 0.05 | 30.84 $\pm$ 0.53 | 80.49 $\pm$ 1.45
> > > |  | MIT-L ($\Delta \lambda > 0.5$) | 1 | 77.14 $\pm$ 0.71 | 2.13 $\pm$ 0.17 | 31.74 $\pm$ 1.11 | 82.87 $\pm$ 3.24
> > > |  | Kernel Warping Mixup (Ours) | 1 | 79.62 $\pm$ 0.68 | 1.84 $\pm$ 0.22 | 29.18 $\pm$ 0.78 | 80.46 $\pm$ 2.08

---

> > > > ### Author Response · Authors · 2023-11-21
> > > > **Table of results 2/2**
> > > >
> > > > ### Table 2: Empirical study of the effect of distance when mixing using Resnet34 on C10 and C100
> > > > Dataset | Quantile of distance | Accuracy | ECE | Brier | NLL |
> > > > |---|---|:---:|:---:|:---:|:---:|
> > > > | C10 | Lower 0.0 / Higher 1.0 (ERM Baseline) | 94.69 $\pm$ 0.27 | 0.82 $\pm$ 0.11 | 8.07 $\pm$ 0.31 | 17.50 $\pm$ 0.61 |
> > > > | | Lower 1.0 / Higher 0.0 (Mixup) | 95.97 $\pm$ 0.27 | 1.36 $\pm$ 0.13 | 6.53 $\pm$ 0.36 | 16.35 $\pm$ 0.72 |
> > > > | | Lower 0.1 | 95.59 $\pm$ 0.42 | 0.878 $\pm$ 0.255 | 7.20 $\pm$ 0.56 | 16.56 $\pm$ 1.52 |
> > > > | | Lower 0.25 | 95.73 $\pm$ 0.18| 1.74 $\pm$ 0.45 | 7.07 $\pm$ 0.26 | 19.39 $\pm$ 1.11 |
> > > > | | Lower 0.5 | 95.88 $\pm$ 0.28 | 1.56 $\pm$ 0.28 | 6.68 $\pm$ 0.34 | 15.86 $\pm$ 0.71 |
> > > > | | Lower 0.75 | 96.16 $\pm$ 0.09 | 1.12 $\pm$ 0.16 | 6.35 $\pm$ 0.15 | 15.20 $\pm$ 0.44 |
> > > > | | Lower 0.9 | 96.31 $\pm$ 0.08 | 1.10 $\pm$ 0.05 | 6.14 $\pm$ 0.11 | 15.16 $\pm$ 0.29 |
> > > > | | Higher 0.9 | 95.58 $\pm$ 0.34 | 1.86 $\pm$ 0.25 | 7.4 $\pm$ 0.48 | 20.32 $\pm$ 1.25 |
> > > > | | Higher 0.75 | 95.91 $\pm$ 0.14 | 1.85 $\pm$ 0.17 | 6.84 $\pm$ 0.22 | 20.06 $\pm$ 1.12 |
> > > > | | Higher 0.5 | 95.58 $\pm$ 0.28 | 1.67 $\pm$ 0.13 | 7.23 $\pm$ 0.37 | 19.12 $\pm$ 0.74 |
> > > > | | Higher 0.25 | 95.98 $\pm$ 0.3 | 1.24 $\pm$ 0.18 | 6.65 $\pm$ 0.51 | 17.06 $\pm$ 0.99 |
> > > > | | Higher 0.1 | 96.28 $\pm$ 0.03 | 1.13 $\pm$ 0.11 | 6.14 $\pm$ 0.04 | 15.24 $\pm$ 0.37 |
> > > > | C100 | Lower 0.0 / Higher 1.0 (ERM Baseline) | 73.47 $\pm$ 1.59 | 2.54 $\pm$ 0.15 | 36.47 $\pm$ 2.05 | 100.82 $\pm$ 6.93 |
> > > > | | Lower 1.0 / Higher 0.0 (Mixup) | 78.11 $\pm$ 0.57 | 2.49 $\pm$ 0.19 | 31.06 $\pm$ 0.69 | 87.94 $\pm$ 1.98 |
> > > > | | Lower 0.1 | 75.40 $\pm$ 0.53 | 3.48 $\pm$ 0.24 | 35.92 $\pm$ 0.5 | 105.87 $\pm$ 1.41 |
> > > > | | Lower 0.25 | 77.14 $\pm$ 0.51 | 2.54 $\pm$ 0.22 | 32.95 $\pm$ 0.62 | 95.42 $\pm$ 1.78 |
> > > > | | Lower 0.5 | 77.66 $\pm$ 0.15 | 1.85 $\pm$ 0.43 | 31.94 $\pm$ 0.28 | 89.97 $\pm$ 1.53 |
> > > > | | Lower 0.75 | 78.43 $\pm$ 0.62 | 1.95 $\pm$ 0.6 | 30.64 $\pm$ 0.7 | 85.06 $\pm$ 1.89 |
> > > > | | Lower 0.9 | 79.24 $\pm$ 0.7 | 1.99 $\pm$ 0.03 | 29.72 $\pm$ 0.94 | 82.54 $\pm$ 2.82 |
> > > > | | Higher 0.9 | 77.3 $\pm$ 0.43 | 1.92 $\pm$ 0.22 | 32.0 $\pm$ 0.59 | 88.69 $\pm$ 1.67 |
> > > > | | Higher 0.75 | 77.8 $\pm$ 1.05 | 2.29 $\pm$ 0.24 | 31.48 $\pm$ 1.18 | 88.16 $\pm$ 3.86 |
> > > > | | Higher 0.5 | 78.74 $\pm$ 0.43 | 2.52 $\pm$ 0.22 | 30.37 $\pm$ 0.56 | 84.64 $\pm$ 1.63 |
> > > > | | Higher 0.25 | 78.51 $\pm$ 84.64 | 2.34 $\pm$ 0.26 | 30.42 $\pm$ 0.59 | 84.64 $\pm$ 2.23 |
> > > > | | Higher 0.1 | 79.14 $\pm$ 0.53 | 2.23 $\pm$ 0.34 | 29.62 $\pm$ 0.51 | 82.22 $\pm$ 1.28 |
> > > >
> > > > ### Table 3: Performance and calibration comparison when using global statistics instead of batch statistics for Airfoil and Exchange Rate
> > > >
> > > > | Dataset | Methods | $\alpha$ | RMSE | MAPE | UCE | ENCE |
> > > > | --- | --- | :---: | :---: | :---: | :---: | :---: |
> > > > | Airfoil | ERM Baseline | - | 2.843 $\pm$ 0.311 | 1.720 $\pm$ 0.219 | 107.6 $\pm$ 19.179 | 0.0210 $\pm$ 0.0078
> > > > | | Mixup | 0.5 | 3.311 $\pm$ 0.207 | 2.003 $\pm$ 0.126 | 147.1 $\pm$ 33.979 | 0.0212 $\pm$ 0.0063
> > > > | | Manifold Mixup | 0.5 | 3.230 $\pm$ 0.177 | 1.964 $\pm$ 0.111 | 126.0 $\pm$ 15.759 | 0.0206 $\pm$ 0.0064
> > > > |  | C-Mixup | 0.5 | 2.850 $\pm$ 0.13 | 1.706 $\pm$ 0.104 | 111.235 $\pm$ 32.567 | 0.0190 $\pm$ 0.0075
> > > > |  | Batch - Kernel Warping Mixup (Ours) | 0.5 | 2.807 $\pm$ 0.261 | 1.694 $\pm$ 0.176 | 126.0 $\pm$ 23.320 | 0.0180 $\pm$ 0.0047
> > > > |  | Global - Kernel Warping Mixup (Ours) | 0.5 | 2.568 $\pm$ 0.235 | 1.529 $\pm$ 0.143 | 95.55 $\pm$ 14.737 | 0.0155 $\pm$ 0.0046
> > > > | Exch. Rate | ERM Baseline | - | 0.019 $\pm$ 0.0024 | 1.924 $\pm$ 0.287 | 0.0082 $\pm$ 0.0028 | 0.0364 $\pm$ 0.0074
> > > > |  | Mixup | 1.5 | 0.0192 $\pm$ 0.0025 | 1.926 $\pm$ 0.284 | 0.0074 $\pm$ 0.0022 | 0.0352 $\pm$ 0.0059
> > > > |  | Manifold Mixup | 1.5 | 0.0196 $\pm$ 0.0026 | 2.006 $\pm$ 0.346 | 0.0086 $\pm$ 0.0029 | 0.0382 $\pm$ 0.0085
> > > > |  | C-Mixup | 1.5 | 0.0188 $\pm$ 0.0017 | 1.893 $\pm$ 0.222 | 0.0078 $\pm$ 0.0020 | 0.0360 $\pm$ 0.0064
> > > > |  | Batch - Kernel Warping Mixup (Ours) | 1.5 | 0.0186 $\pm$ 0.0020 | 1.872 $\pm$ 0.235 | 0.0074 $\pm$ 0.0019 | 0.0346 $\pm$ 0.0050
> > > > |  | Global - Kernel Warping Mixup (Ours) | 1.5 | 0.0186 $\pm$ 0.0020 | 1.875 $\pm$ 0.236 | 0.0074 $\pm$ 0.0019 | 0.0346 $\pm$ 0.0050

---

> > ### Comment · Reviewer_djAy · 2023-11-23
> >
> > The authors have carefully answered my questions, supplementing their explanations with additional experiments. Given my limited expertise in Calibration, it is hard for me to access the contribution in the context thoroughly. Consequently, I would like to raise my score to 6; however, my confidence has decreased to 3.

---

### Official Review · Reviewer_2Wwo · 2023-10-31

**Soundness:** 2 fair
**Presentation:** 3 good
**Contribution:** 2 fair
**Rating:** 5
**Confidence:** 3

**Summary:**

This paper aims to improve the mixup algorithm by improving its data interpolation policy. Particularly, the paper proposes Kernel Warping Mixup that can dynamically change the sampling distribution of interpolation coefficient $\lambda$, so that when mixing data ponts that are "closer" under a certain metrics, the choice of $\lambda$ can have a higher degree of freedom; when mixing data points that are "farther", $\lambda$ should be chosen closer to 0 or 1. Experiments on both classification tasks and regression tasks are conducted, and the results show that the proposed Kernel Warping Mixup improves both the test accuracy and the calibration compared with conventional Mixup.

**Strengths:**

1. Proposed a variant mixup algorithm that has shown improvement on both generalization and calibration compared to conventional mixup.

2. Adequate experiments on common datasets, both of classifrication tasks and regression tasks.

3. The thought process of designing the proposed algorithm is explained clearly.

**Weaknesses:**

1. The idea of dynamically controlling the sampling of $\lambda$ in mixup is not novel, and the idea of controlling the pairing of the data examples based on distributional similarities (like k-mixup) is also well-investigated. As a result, combining these two types of ideas to formulate a new algorithm, like the one proposed in this paper, is also intuitively and empirically straightforward, and doesn't seem to be much surprising or interesting. In my opinion it the amount of contributions in this work is not sufficient to be a full paper in ICLR.

2. As a experiment-based work, the datasets used in the experiments are not adequate. For example, in classifications the authors only investigated their proposed algorithm on image datasets. In fact, some datasets, especially the ones with lower dimensionalities, tend to benefit less or even negatively from mixup, and also L2 distance between data points in these datasets may be more statistically meaningful. Such datasets are also worth of experiment verifications of the Kernel Warping Mixup. This also applies to the regression tasks.

3. The essential hyperparameters $\tau_{max}$ and $\tau_{std}$ used are chosen through cross-validation to find the optimal values. This may cost extra time before the real training is even started.

**Questions:**

1. In the last paragraph of Section 1, "... improve both performance and ...", what performance exactly? Is it refering to generalization performance?

2. Section 3.3. Why $\tau$ should be exponentially correlated with the distance?

3. Is there any principle or strategy to select the metrics of similarity? Like L2 or optimal transports or else?

4. How is the cross-validation used to find the hyperparameters conducted in details? Is it conducted before the real training? Or is it conducted simultaneously during the training? What objective is considered in finding the "optimal" $\tau$'s?

5. Algorithm 1. It seems that the values of $\tau$'s for the inputs and the targets are computed in the identical way. Then what is the point of defining them separately? And also, if the $\tau^o$ is computed separately, how would one define the similarity between two one-hot labels in classifications? Is it going to be like a simple equity indicator function?

6. Section 4.2. Why is the input distance or embedding distance not taken into consideration here? For some regression problem, I believe the similarity between two targets doesn't necessarily indicate a comparable similarity between their inputs.

---

> ### Author Response · Authors · 2023-11-21
> **Response to Reviewer 2Wwo**
>
> We thank the reviewer for taking time to review our work and for their constructive feedback.
>
> - (W1) May we ask the reviewer for references linked to the claim on the novelty of our work ? As far as we know, at the time of submission, we did not find related work that also proposed to dynamically change the underlying distribution of $\lambda$ during training. Furthermore, we are not controlling the *pairing of points*, but how strongly the data are mixed by changing the sampling distribution of the $\lambda$, using the **distance between the two points** to mix, and **not distribution similarities unlike k-mixup** (Greenewald et al., 2021).
> - (W2) Our goal is to study the impact of mixup on calibration in general for both classification and regression tasks. We considered datasets for classification that are widely used in the literature to accurately compare with other baselines, and they are mainly image datasets. However, it is not the case for the *regression tasks*, where the datasets used are *tabular* and *times series* data. Specifically, on *Airfoil* (tabular data), we can see that **both mixup and manifold mixup have negative effects** on performance and calibration, but not with neither C-Mixup nor our approach. In this case, we are using L2 distance between labels since it works best in this case as shown in Yao et al. (2022).
> - (W3) While we agree that finding the correct values for $\tau_{max}$ and $\tau_{std}$ through cross validation beforehand takes additional time, we would like to point that **it is the case for any hyperparameter** and that **other mixup methods also have additional hyperparameters**, such as $\alpha$ in the Beta distributions or the strength of the regularization in RegMixup, or the layers selected to apply mixup in Manifold Mixup. In our case, **we do not have to search for a good potential $\alpha$** since it is already reflected in both $\tau_{max}$ and $\tau_{std}$, which makes one less hyperparameter.
> - (Q1) When stating that our method "improves both performance and calibration", we are referring to generalization performance in-distribution, i.e. accuracy on the test set, *and* that we improve calibration at the same time.
> - (Q2) **The exponential correlation is linked to the warping function that we consider**, i.e. in our case, the Beta CDF function. As can be seen in figure 2 of the paper, there is a *logarithmic* correlation on the shape of the Beta CDF and the parameter $\tau$. Based on this, we defined the similarity kernel with an *exponential correlation* with the distance. As mentioned in the paper, the correlation with the distance will depend on the warping function considered in the framework.
> - (Q3) We present in the paper several possible distance metrics that we tried, L2 distance between inputs, between embeddings of input, between classification weights of the associated true class or between labels. We also tried Cosine distance instead of L2 distance, but did not observe meaningful difference between the two. However, we agree that one could consider optimal transport distances as well *if each data can be seen as a distribution* (since we are computing similarity for a pair of data), which could be possible for images using distributions of pixel values.
> - (Q4) The cross-validation is conducted before training, with a stratified sampling on a 90/10 split of the training set, and the results are averaged across 4 different splits. The hyperparameters are selected to have **a good trade-off between calibration and accuracy**. As mentioned in Pinto et al. (2022), *cross-validating hyperparameters based solely on the ECE can prefer models with lower accuracy but better calibration*. However, a method improving calibration should avoid degrading accuracy. We will clarify the process of the cross validation in the paper.
> - (Q5) We thank the reviewer for pointing out this typo in the algorithm. We compute $\tau_i^{(i)}$ with $\tau^{(i)}(\mathbf{x}, i, \sigma; \tau^{(i)}\_{max}, \tau^{(i)}\_{std})$, and $\tau_i^{(o)}$ with $\tau^{(o)}(\mathbf{x}, i, \sigma; \tau^{(o)}\_{max}, \tau^{(o)}\_{std})$. If one wants to use different $\tau^{(i)}$ and $\tau^{(o)}$, one has to use different pairs of $(\tau^{(i)}\_{max}, \tau^{(i)}\_{std})$ and $(\tau^{(o)}\_{max}, \tau^{(o)}\_{std})$.
> - (Q6) While we agree that we could have considered input or embedding distance for regression tasks, it was shown in Yao et al. (2022), that label distance works best in this case. We thank the reviewer for this remark and will make it clear in the revised version.
>
> ### References
>
> - Pinto et al., RegMixup: Mixup as a Regularizer Can Surprisingly Improve Accuracy and Out Distribution Robustness. NeurIPS, 2022
> - Yao et al., C-Mixup: Improving generalization in regression. NeurIPS, 2022.
> - Greenewald et al., k-Mixup Regularization for Deep Learning via Optimal Transport. arXiv:2106.02933, 2021

---

> > ### Comment · Reviewer_2Wwo · 2023-11-22
> > **Confirmation of the response**
> >
> > Thanks for the detailed response. For (W1), this paper[1] can be a reference in which the authors investigate a new algorithm to adaptively learn the mixing policy from the training data.
> >
> > Also for (Q5), it's still unclear to me how the similarity between two one-hot labels can be defined or computed.
> >
> > [1]Guo et al., MixUp as Locally Linear Out-Of-Manifold Regularization. arXiv:1809.02499v3

---

> > > ### Author Response · Authors · 2023-11-22
> > >
> > > Thank you for the quick answer.
> > > - The AdaMixup method discussed in the provided paper differs from our proposed approach in several crucial ways:
> > >   1. AdaMixup is based on **two additional networks**. The first one predicts if the mixed sample is in the manifold of data, for regularization. The second one predicts a range of mixing values. **The method includes a lot more total parameters** than ours and **the training is more complex** with several regularization tasks in parallel.
> > >   2. **The total loss includes training on non-mixed samples**, similarly to RegMixup, which doubles the batch size in practice and different to both the classical Mixup and our approach.
> > >   3. The mixing policy predicted is very different to what we use. The policy is an interval ($\alpha$, $\alpha + \Delta$), and **the predicted range of values is very narrow around 0.5** which reduces a lot the diversity of mixed samples obtained.
> > >
> > > In general, we propose a much more efficient and simple way to adapt the mixing coefficients to the data. We mentioned this paper in the related work section, but we will include this discussion. Thank you for the reference.
> > > - Regarding (Q5), **we do not compute similarity between one-hot labels**. As presented in the paper and in the answer to (Q3), we considered three different distance metrics in classification tasks: *L2 distance on inputs*, *L2 distance on embeddings of inputs*, and *L2 distance between class weights* of the classification layer associated to the input data.

---

> > > > ### Comment · Reviewer_2Wwo · 2023-11-22
> > > >
> > > > Page 7: "In Table 1, we compared results for each of these choice and for different combinations of similarity
> > > > between inputs and targets." Here you have mentioned similarity between inputs and targets, and you have also indicated "output similarity" in the following sentence.", now it turns out you don't intend to compute the distance or similarity between the actual "labels" or "model outputs", I assume there is an abuse of terminology here? Or I would suggest that you clarify the concept of "target similarity" or "output similarity" in this context. For example, in Table 1, under the column of "output similarity" there is a "input (distance)", this will cause some misunderstandings

---

### Official Review · Reviewer_YFeL · 2023-11-02

**Soundness:** 3 good
**Presentation:** 4 excellent
**Contribution:** 3 good
**Rating:** 8
**Confidence:** 3

**Summary:**

This paper generally contributes a new way to data augmentation by change distributions of training data.

Here is a general summary:
1. Authors define warping functions to change the underlying distributions used for sampling
interpolation coefficients. This defines a general framework that allows to disentangle inputs
and labels when mixing, and spans several variants of mixup.
2. Authors proposed to then apply a similarity kernel that takes into account the distance between
points to select a parameter for the warping function tailored to each pair of points to mix,
governing its shape and strength. This tailored function warps the interpolation coefficients
to make them stronger for similar points and weaker otherwise.
3. Authors show that our Kernel Warping Mixup is general enough to be applied in classification as
well as regression tasks.

This major contribution of this paper is mixing the idea of regularization and kernel function and try to
solve the problem from data perspective. If experiments result is convincible, this is a new way to consider
data augmentation.

**Strengths:**

The ideology of this paper is quite plausible and is theoretically robust.

**Weaknesses:**

I don't think regression task can be convincing as downstream task for novelty. Regression task, in somehow, can already achieve very good result. I think a challenging downstream task such as object detection can make this paper more attractive.

**Questions:**

How do you define/calculate distance?

---

> ### Author Response · Authors · 2023-11-21
> **Response to Reviewer YFeL**
>
> We thank the reviewer for their kind comments on our work.
>
> - (W1) The main goal of the paper is to study and improve the impact of mixup on calibration in general. We present experiments for both classification and regression tasks, since it is **two very different settings**, and not that well studied in regression. Nevertheless, we show the *flexibility* of our framework, which can improve calibration in both tasks. However, as mentioned in the conclusion of our paper, we are interested to apply our method on more difficult regression tasks such as Monocular Depth Estimation, but leave that to future work.
> - (Q1) As presented in Section 3.3, we use an L2 distance, on either input data, embedding of inputs, classification weights or labels depending on the task.

---

### Official Review · Reviewer_iaPi · 2023-11-05

**Soundness:** 2 fair
**Presentation:** 3 good
**Contribution:** 2 fair
**Rating:** 3
**Confidence:** 5

**Summary:**

Mixup data augmentations are a widely used technique for deep learning, and most methods are focused on selecting the right points to mix or designing favorite mixing strategies. This paper tries to improve mixup by mixing similar data points more frequently than less similar ones and proposes a dynamically changing underlying distribution of mixing interpolation coefficients through warping functions, dubbed Kernel Warping Mixup. Extensive experiments for classification and regression tasks demonstrate the effectiveness and generalization abilities of the proposed mixup.

**Strengths:**

* (S1) It is an interesting and novel perspective of improving mixup augmentations by mixing more similar samples than less similar ones (but it also needs verification and empirical analysis to ensure the importance, as mentioned in W1). Experiment results show the effectiveness of the proposed method in comparison to classical mixup variants on both classification and regression tasks.

* (S2) Various analyzing metrics are used to verify the effectiveness of mixup classification and regression tasks, e.g., ECE, UCE, and NLL, which are not well studied in previous works.

* (S3) The overall writing is fruentcent and easy to follow. The implementation details and source code are available.

**Weaknesses:**

* (W1) Is the studied problem really important to mixup augmentations, i.e., it is practically useful to conduct mixup interpolation with similar samples? I cannot find any empirical analysis demonstrating that previous mixup methods will encounter serious drawbacks (e.g., performances, calibration abilities, generalization abilities to tasks) because of not mixing similar samples well. For example, the authors should provide some visualizations of effects or statistics to demonstrate the problem in addition to Figure 1.

* (W2) Weak experiments. In comparison to recently published works on mixup augmentation, this paper lacks comprehensive and solid comparison experiments to verify the effectiveness from three aspects. (a) The compared baselines are restricted to classical methods in classification tasks, and the performance gains are limited in all comparison results. These results make me doubt the importance of the studied problem in this paper, i.e., mixing more similar mixup samples than less similar ones. (b) The experiments are small-scale (e.g., CIFAR-10/100) with classical network architectures (ResNet variants) and old-fashioned baselines. There are many open-source mixup methods and benchmarks for classification and regression tasks [1, 2], and I suggest the authors consider more practical and modern experiment settings (e.g., large-scale experiments on ImageNet, modern Transformer backbones on CIFAR-100). (c) Since the proposed method is orthogonal to these mixup methods that improve mixup policies of samples (e.g., CutMix variants [3, 4], or randomly combining Mixup and CutMix [5]) or labels (e.g., TransMix [6], Decouple Mixup [7], and MixupE [8]), the authors should verify whether the proposed method can improve these existing mixup algorithms, rather than only test upon the vanilla Input Mixup.

* (W3) Hyper-parameter sensitivity. As shown in Appendix D and E (Table 6), the hyper-parameters of the proposed method vary significantly on different datasets and tasks. The ablation and sensitivity analysis of these hyper-parameters should be added. Meanwhile, I wonder how the authors determine the two hyper-parameters, which should be detailed in the appendix.

* (W4) The related work section is not well presented. I suggest the author combine Sec. 2.2 and Sec. 3.1 as a new section called Preliminary. Meanwhile, the authors may include more recently published mixup algorithms in different categories.

### Reference
[1] OpenMixup: A Comprehensive Mixup Benchmark for Visual Classification. arXiv, 2022.

[2] C-Mixup: Improving Generalization in Regression. NeurIPS, 2022.

[3] CutMix: Regularization Strategy to Train Strong Classifiers with Localizable Features. ICCV, 2019.

[4] PuzzleMix: Exploiting Saliency and Local Statistics for Optimal Mixup. ICML, 2020.

[5] Training data-efficient image transformers & distillation through attention. ICML, 2021.

[6] TransMix: Attend to Mix for Vision Transformers. CVPR, 2022.

[7] Harnessing Hard Mixed Samples with Decoupled Regularizer. NeurIPS, 2023.

[8] MixupE: Understanding and Improving Mixup from Directional Derivative Perspective. UAI, 2023.

**Questions:**

Please refer to the weaknesses I mentioned.

---

> ### Author Response · Authors · 2023-11-21
> **Response to Reviewer iaPi**
>
> We thank the reviewer for their review and extensive feedback on our work.
>
> - (W1) The assumption underlying this work is that there is a trade-off in data augmentation procedures between adding *diversity* and introducing *uncertainty* through **manifold intrusion** (Guo et al., 2019; Baena et al., 2022), *i.e*, conflicts between the synthetic labels of the mixed-up examples and the labels of original training data. To show that, we conduct in Table 1 an empirical analysis that compares *accuracy* and *calibration* metrics when mixing *only* pairs of points with distances lower or higher than a given quantile of the overall pairwise distances within the batch, using a Resnet34 on CIFAR10 and CIFAR100 datasets. Note that one should compare results of "Lower $q$" with "Higher $1 - q$" to have equivalent numbers of possible element to mix with (*diversity*). We can see that, in general, **mixing pairs with lower distances leads to better calibration than mixing pairs with higher distances** (*uncertainty*). The introduction of uncertainty when mixing pairs with high distance can be explained by *manifold intrusion* (Guo et al., 2019; Baena et al., 2022). These results confirm that there is a trade-off between adding *diversity* and *uncertainty with* data augmentation.
> - (W2) Along with the empirical analysis detailed above, we also provide experiments on the Tiny-Imagenet benchmark in Table 2. We can see that our approach improves both performance and calibration over Mixup baselines on this dataset as well, and reach similar calibration than RegMixup. We are working on adding the other baselines for this dataset as well. As mentioned in the conclusion of the paper, we are working on combining our approach with other mixup methods. Regarding the references mentioned by the reviewer:
>     - The OpenMixup paper [1] and library mentioned is still under construction, but **we plan to contribute to this library** once our paper is accepted.
>     - **We are already comparing against C-Mixup [2]** on the same datasets in the experiments on regression tasks.
>     - CutMix [3,5], PuzzleMix [4] and TransMix [6] are **methods focused on image data**. As our main goal is to study and improve the impact of mixup on *calibration in general*, we do not want to restrict our evaluation to image datasets and image methods. We consider regression tasks on tabular and time series data, to evaluate calibration of Mixup methods in this setting, which have rarely been investigated in the literature, but all of the above methods are not adapted to these types of data.
>     - As both MixupE [8] and Decouple Mixup [7] have been published **very recently**, implementing these methods will require further work.
> - (W3) *As mentioned in Section 4.1*, we search for the best hyperparameters through **cross-validation**. The cross-validation is conducted before training, with a stratified sampling on a 90/10 split of the training set, and the results are averaged across 4 different splits. The optimal values will vary between datasets, since they depend on the statistics of the pairwise distances. We assume that the behavior of these parameters is also impacted by the number of class and their separability in each dataset. We provide the resulting heatmaps of the cross validation in Figure 5 and Figure 6 for a better visualization, but we will also include the exact values obtained for all the cross validations in the Appendix.
> - (W4) We will make sure to add the latest relevant papers that we might have missed since the moment of submission. Regarding the two subsections mentioned by the reviewer, we will try to combine them, but it might be difficult since they cover different topics. While Sec. 2.2 presents related work on calibration, Sec. 3.1 introduces notations used throughout the paper.

---

> > ### Author Response · Authors · 2023-11-21
> > **Tables of results**
> >
> > ### Table 1: Empirical study of the effect of distance when mixing using Resnet34 on C10 and C100
> > Dataset | Quantile of distance | Accuracy | ECE | Brier | NLL |
> > |---|---|:---:|:---:|:---:|:---:|
> > | C10 | Lower 0.0 / Higher 1.0 (ERM Baseline) | 94.69 $\pm$ 0.27 | 0.82 $\pm$ 0.11 | 8.07 $\pm$ 0.31 | 17.50 $\pm$ 0.61 |
> > | | Lower 1.0 / Higher 0.0 (Mixup) | 95.97 $\pm$ 0.27 | 1.36 $\pm$ 0.13 | 6.53 $\pm$ 0.36 | 16.35 $\pm$ 0.72 |
> > | | Lower 0.1 | 95.59 $\pm$ 0.42 | 0.878 $\pm$ 0.255 | 7.20 $\pm$ 0.56 | 16.56 $\pm$ 1.52 |
> > | | Lower 0.25 | 95.73 $\pm$ 0.18| 1.74 $\pm$ 0.45 | 7.07 $\pm$ 0.26 | 19.39 $\pm$ 1.11 |
> > | | Lower 0.5 | 95.88 $\pm$ 0.28 | 1.56 $\pm$ 0.28 | 6.68 $\pm$ 0.34 | 15.86 $\pm$ 0.71 |
> > | | Lower 0.75 | 96.16 $\pm$ 0.09 | 1.12 $\pm$ 0.16 | 6.35 $\pm$ 0.15 | 15.20 $\pm$ 0.44 |
> > | | Lower 0.9 | 96.31 $\pm$ 0.08 | 1.10 $\pm$ 0.05 | 6.14 $\pm$ 0.11 | 15.16 $\pm$ 0.29 |
> > | | Higher 0.9 | 95.58 $\pm$ 0.34 | 1.86 $\pm$ 0.25 | 7.4 $\pm$ 0.48 | 20.32 $\pm$ 1.25 |
> > | | Higher 0.75 | 95.91 $\pm$ 0.14 | 1.85 $\pm$ 0.17 | 6.84 $\pm$ 0.22 | 20.06 $\pm$ 1.12 |
> > | | Higher 0.5 | 95.58 $\pm$ 0.28 | 1.67 $\pm$ 0.13 | 7.23 $\pm$ 0.37 | 19.12 $\pm$ 0.74 |
> > | | Higher 0.25 | 95.98 $\pm$ 0.3 | 1.24 $\pm$ 0.18 | 6.65 $\pm$ 0.51 | 17.06 $\pm$ 0.99 |
> > | | Higher 0.1 | 96.28 $\pm$ 0.03 | 1.13 $\pm$ 0.11 | 6.14 $\pm$ 0.04 | 15.24 $\pm$ 0.37 |
> > | C100 | Lower 0.0 / Higher 1.0 (ERM Baseline) | 73.47 $\pm$ 1.59 | 2.54 $\pm$ 0.15 | 36.47 $\pm$ 2.05 | 100.82 $\pm$ 6.93 |
> > | | Lower 1.0 / Higher 0.0 (Mixup) | 78.11 $\pm$ 0.57 | 2.49 $\pm$ 0.19 | 31.06 $\pm$ 0.69 | 87.94 $\pm$ 1.98 |
> > | | Lower 0.1 | 75.40 $\pm$ 0.53 | 3.48 $\pm$ 0.24 | 35.92 $\pm$ 0.5 | 105.87 $\pm$ 1.41 |
> > | | Lower 0.25 | 77.14 $\pm$ 0.51 | 2.54 $\pm$ 0.22 | 32.95 $\pm$ 0.62 | 95.42 $\pm$ 1.78 |
> > | | Lower 0.5 | 77.66 $\pm$ 0.15 | 1.85 $\pm$ 0.43 | 31.94 $\pm$ 0.28 | 89.97 $\pm$ 1.53 |
> > | | Lower 0.75 | 78.43 $\pm$ 0.62 | 1.95 $\pm$ 0.6 | 30.64 $\pm$ 0.7 | 85.06 $\pm$ 1.89 |
> > | | Lower 0.9 | 79.24 $\pm$ 0.7 | 1.99 $\pm$ 0.03 | 29.72 $\pm$ 0.94 | 82.54 $\pm$ 2.82 |
> > | | Higher 0.9 | 77.3 $\pm$ 0.43 | 1.92 $\pm$ 0.22 | 32.0 $\pm$ 0.59 | 88.69 $\pm$ 1.67 |
> > | | Higher 0.75 | 77.8 $\pm$ 1.05 | 2.29 $\pm$ 0.24 | 31.48 $\pm$ 1.18 | 88.16 $\pm$ 3.86 |
> > | | Higher 0.5 | 78.74 $\pm$ 0.43 | 2.52 $\pm$ 0.22 | 30.37 $\pm$ 0.56 | 84.64 $\pm$ 1.63 |
> > | | Higher 0.25 | 78.51 $\pm$ 84.64 | 2.34 $\pm$ 0.26 | 30.42 $\pm$ 0.59 | 84.64 $\pm$ 2.23 |
> > | | Higher 0.1 | 79.14 $\pm$ 0.53 | 2.23 $\pm$ 0.34 | 29.62 $\pm$ 0.51 | 82.22 $\pm$ 1.28 |
> >
> > ### Table 2: Performance and calibration comparison with Resnet50 on Tiny-Imagenet
> >
> > | Method | $\alpha$ | Accuracy | ECE | Brier | NLL |
> > |---|:---:|:---:|:---:|:---:|:---:|
> > | ERM Baseline | - | 66.74 $\pm$ 0.34 | 1.62 $\pm$ 0.22 | 1.36 $\pm$ 0.02 | 0.44 $\pm$ 0.0045 |
> > | Mixup | 1 | 67.21 $\pm$ 0.21 | 1.63 $\pm$ 0.10 | 1.37 $\pm$ 0.01 | 0.44 $\pm$ 0.0028 |
> > | | 0.5 | 67.34 $\pm$ 0.69 | 1.56 $\pm$ 0.05 | 1.36 $\pm$ 0.04 | 0.44 $\pm$ 0.0099 |
> > | | 0.1 | 66.48 $\pm$ 0.57 | 1.66 $\pm$ 0.17 | 1.42 $\pm$ 0.02 | 0.45 $\pm$ 0.0061 |
> > | MixupIO | 1 | 66.17 $\pm$ 0.28 | 1.49 $\pm$ 0.21 | 1.37 $\pm$ 0.01 | 0.45 $\pm$ 0.0033 |
> > | | 0.5 | 66.98 $\pm$ 0.39 | 1.75 $\pm$ 0.12 | 1.35 $\pm$ 0.01 | 0.44 $\pm$ 0.0025 |
> > | | 0.1 | 65.87 $\pm$ 0.57 | 1.51 $\pm$ 0.21 | 1.39 $\pm$ 0.02 | 0.45 $\pm$ 0.0057 |
> > | RegMixup | 10 | 69.39 $\pm$ 0.62 | 1.31 $\pm$ 0.08 | 1.27 $\pm$ 0.03 | 0.42 $\pm$ 0.0078 |
> > | RegMixup | 20 | 69.71 $\pm$ 0.42 | 1.17 $\pm$ 0.19 | 1.24 $\pm$ 0.03 | 0.41 $\pm$ 0.0067 |
> > | Kernel Warping Mixup (Ours) | 1 | 68.18 $\pm$ 0.26 | 1.29 $\pm$ 0.35 | 1.33 $\pm$ 0.02 | 0.43 $\pm$ 0.0046 |

---

> ### Comment · Reviewer_iaPi · 2023-11-23
> **Reply to Rebuttal Feedback**
>
> Thanks for the detailed response to my concerns. Sorry for the late reply at the end of the rebuttal period! My concerns about W3 and W4 have been addressed (it is better for the authors to update them in the revision). However, I am not convinced by the replies to the first two issues, and I decided to maintain my rating at the current stage.
>
> * (W1) Despite the authors explaining the motivation and the main problem in this paper, I think some empirical studies are required to demonstrate the studied problem really existed in previous works. This is directly related to the effectiveness of the proposed method. Since the performance gains of the proposed method are not significant or even worse than existing methods,  I really need the relevant evidence to support the problems and statements in the authors' response.
>
> * (W2) I have acknowledged the purpose of the proposed methods and the reasons for not conducting the comparison experiments I recommended. This leaves me wondering what is the practical value of the proposed approach if it mainly addresses small-scale image data, time series data, and tabular data. From my perspective, it makes this article's contribution seem inadequate to be accepted by the ICLR community.
>
> Except for the unsolved issues, I appreciate the improved perspective of this article, the comprehensive evaluation metrics, and the overall presentations.

---

> ### Author Response · Authors · 2023-11-23
>
> Thank you for taking the time to reply.
>
> (W1)
>    1. **Empirical evidence of the impact of distance on calibration:** We have conducted the empirical studies mentioned by the reviewer, in Table 1 of our first answer, to precisely study the impact on calibration of the distance between the points to mix. These experiments bring empirical evidence that, indeed, **mixing pairs with lower distances leads to better calibration than mixing pairs with higher distances**.
>    2. **On the importance of the contribution:** The problem of the trade-off between performance and calibration with Mixup have been **extensively studied in previous work** (Thulasidasan et al. 2019; Zhang et al., 2022; Wang et al., 2023), as mentioned in Section 2.2. The motivation of our paper is to propose **a more efficient approach to achieve a better trade-off in terms of performance and calibration** by controlling the impact of distance on mixing.
>
> (W2) We would like to point that what the reviewer calls "small-scale image data, time series data and tabular data", encompasses a lot of possible applications and represents a wider scope than previous Mixup papers limited to image data. During the rebuttal, we have also included experiments on Tiny-ImageNet, which is on a bigger scale than CIFAR. Nothing prevents applying our approach to larger scale, this is something that we are working on, but needs more time.
>
> The paper is now updated with the changes.
>
> References:
> - Thulasidasan et al., On mixup training: Improved calibration and predictive uncertainty for deep neural networks. NeurIPS 2019
> - Zhang et al., When and how mixup improves calibration. ICML 2022
> - Wang et al., On the pitfall of mixup for uncertainty calibration. CVPR 2023

---

### Author Response · Authors · 2023-11-21
**General Answer**

We thank all the reviewers for taking the time to review our work and for their constructive feedbacks. We are glad to hear that our paper is:
- Interesting and with a novel perspective on mixup data augmentation (iaPi, YFeL, djAY)
- easy to follow and explained clearly (iaPi, 2Wwo)

In addition to individual responses, we would like to clarify the following points:
- The main goal of the paper is to study and improve the impact of mixup on *calibration in general* while preserving performance. We experiment on both classification and regression benchmarks, the latter being rarely investigated in the context of mixup and calibration.
- The assumption underlying this work is that there is a trade-off in data augmentation procedures between adding *diversity* and introducing *uncertainty*. While we agree that it is important to cover more regions of the space, it is important to do it without adding errors and avoid *manifold intrusion* (Guo et al., 2019; Baena et al., 2022), *i.e*, conflicts between the synthetic labels of the mixed-up examples and the labels of original training data.
- We propose a flexible and cheap extension of mixup that allows to govern if pairs should be mixed or not depending on pairwise distances, through the underlying distribution of the mixing coefficients. Our method is less expensive in terms of compute and memory than other methods designed for improving calibration (c.f. Appendix B).
- We can consider combining other mixup methods with our proposed one, since the changes between methods are orthogonal to ours. We leave that for future work.

- We are adding three sets of new experiments to answer the questions of the reviewers:
    1. **Table 1: Empirical study of the effect of distance when mixing using Resnet34 on C10 and C100** We present here an experiment aligned with our hypothesis. We compare *accuracy* and *calibration* metrics when mixing *only* pairs of points with distances lower or higher than a given quantile of the overall pairwise distances within the batch, using a Resnet34 on CIFAR10 and CIFAR100 datasets. Note that one should compare results of "Lower $q$" with "Higher $1 - q$" to have equivalent numbers of possible element to mix with (*diversity*). We can see that, in general, *mixing pairs with lower distances leads to better calibration than mixing pairs with higher distances* (*uncertainty*). The introduction of uncertainty when mixing pairs with high distance can be explained by *manifold intrusion* (Guo et al., 2019; Baena et al., 2022). These results confirm that there is a trade-off between adding *diversity* and *uncertainty with* data augmentation.
    2. **Table 2: Results of Manifold Mixup on C10 and C100 with Resnet50** We reproduced results of Manifold Mixup (Verma et al., ICML 2019) on CIFAR 10 and CIFAR100 using a Resnet50. We achieve similar performance improvements with better calibration. As for RegMixup, we can consider combining our method with Manifold Mixup in future work.
    3. **Table 3: Results on Tiny-Imagenet with Resnet50** We provide results on Tiny-Imagenet, a bigger benchmark dataset for image classification. We can see that we improve both performance and calibration over Mixup baselines, and reach similar calibration than RegMixup. We are working on adding the other baselines for this dataset as well.

### References:
- Guo et al., Mixup as locally linear out-of-manifold regularization. AAAI, 2019.
- Baena et al., A local mixup to prevent manifold intrusion. EUSIPCO, 2022.
- Verma et al., Manifold mixup: Better representations by interpolating hidden states. ICML, 2019.

---

> ### Author Response · Authors · 2023-11-21
> **Tables of results 1/2**
>
> ### Table 1: Empirical study of the effect of distance when mixing using Resnet34 on C10 and C100
> Dataset | Quantile of distance | Accuracy | ECE | Brier | NLL |
> |---|---|:---:|:---:|:---:|:---:|
> | C10 | Lower 0.0 / Higher 1.0 (ERM Baseline) | 94.69 $\pm$ 0.27 | 0.82 $\pm$ 0.11 | 8.07 $\pm$ 0.31 | 17.50 $\pm$ 0.61 |
> | | Lower 1.0 / Higher 0.0 (Mixup) | 95.97 $\pm$ 0.27 | 1.36 $\pm$ 0.13 | 6.53 $\pm$ 0.36 | 16.35 $\pm$ 0.72 |
> | | Lower 0.1 | 95.59 $\pm$ 0.42 | 0.878 $\pm$ 0.255 | 7.20 $\pm$ 0.56 | 16.56 $\pm$ 1.52 |
> | | Lower 0.25 | 95.73 $\pm$ 0.18| 1.74 $\pm$ 0.45 | 7.07 $\pm$ 0.26 | 19.39 $\pm$ 1.11 |
> | | Lower 0.5 | 95.88 $\pm$ 0.28 | 1.56 $\pm$ 0.28 | 6.68 $\pm$ 0.34 | 15.86 $\pm$ 0.71 |
> | | Lower 0.75 | 96.16 $\pm$ 0.09 | 1.12 $\pm$ 0.16 | 6.35 $\pm$ 0.15 | 15.20 $\pm$ 0.44 |
> | | Lower 0.9 | 96.31 $\pm$ 0.08 | 1.10 $\pm$ 0.05 | 6.14 $\pm$ 0.11 | 15.16 $\pm$ 0.29 |
> | | Higher 0.9 | 95.58 $\pm$ 0.34 | 1.86 $\pm$ 0.25 | 7.4 $\pm$ 0.48 | 20.32 $\pm$ 1.25 |
> | | Higher 0.75 | 95.91 $\pm$ 0.14 | 1.85 $\pm$ 0.17 | 6.84 $\pm$ 0.22 | 20.06 $\pm$ 1.12 |
> | | Higher 0.5 | 95.58 $\pm$ 0.28 | 1.67 $\pm$ 0.13 | 7.23 $\pm$ 0.37 | 19.12 $\pm$ 0.74 |
> | | Higher 0.25 | 95.98 $\pm$ 0.3 | 1.24 $\pm$ 0.18 | 6.65 $\pm$ 0.51 | 17.06 $\pm$ 0.99 |
> | | Higher 0.1 | 96.28 $\pm$ 0.03 | 1.13 $\pm$ 0.11 | 6.14 $\pm$ 0.04 | 15.24 $\pm$ 0.37 |
> | C100 | Lower 0.0 / Higher 1.0 (ERM Baseline) | 73.47 $\pm$ 1.59 | 2.54 $\pm$ 0.15 | 36.47 $\pm$ 2.05 | 100.82 $\pm$ 6.93 |
> | | Lower 1.0 / Higher 0.0 (Mixup) | 78.11 $\pm$ 0.57 | 2.49 $\pm$ 0.19 | 31.06 $\pm$ 0.69 | 87.94 $\pm$ 1.98 |
> | | Lower 0.1 | 75.40 $\pm$ 0.53 | 3.48 $\pm$ 0.24 | 35.92 $\pm$ 0.5 | 105.87 $\pm$ 1.41 |
> | | Lower 0.25 | 77.14 $\pm$ 0.51 | 2.54 $\pm$ 0.22 | 32.95 $\pm$ 0.62 | 95.42 $\pm$ 1.78 |
> | | Lower 0.5 | 77.66 $\pm$ 0.15 | 1.85 $\pm$ 0.43 | 31.94 $\pm$ 0.28 | 89.97 $\pm$ 1.53 |
> | | Lower 0.75 | 78.43 $\pm$ 0.62 | 1.95 $\pm$ 0.6 | 30.64 $\pm$ 0.7 | 85.06 $\pm$ 1.89 |
> | | Lower 0.9 | 79.24 $\pm$ 0.7 | 1.99 $\pm$ 0.03 | 29.72 $\pm$ 0.94 | 82.54 $\pm$ 2.82 |
> | | Higher 0.9 | 77.3 $\pm$ 0.43 | 1.92 $\pm$ 0.22 | 32.0 $\pm$ 0.59 | 88.69 $\pm$ 1.67 |
> | | Higher 0.75 | 77.8 $\pm$ 1.05 | 2.29 $\pm$ 0.24 | 31.48 $\pm$ 1.18 | 88.16 $\pm$ 3.86 |
> | | Higher 0.5 | 78.74 $\pm$ 0.43 | 2.52 $\pm$ 0.22 | 30.37 $\pm$ 0.56 | 84.64 $\pm$ 1.63 |
> | | Higher 0.25 | 78.51 $\pm$ 84.64 | 2.34 $\pm$ 0.26 | 30.42 $\pm$ 0.59 | 84.64 $\pm$ 2.23 |
> | | Higher 0.1 | 79.14 $\pm$ 0.53 | 2.23 $\pm$ 0.34 | 29.62 $\pm$ 0.51 | 82.22 $\pm$ 1.28 |

---

> > ### Author Response · Authors · 2023-11-21
> > **Tables of results 2/2**
> >
> > ### Table 2: Performance and calibration comparison of Manifold Mixup on C10 and C100 with Resnet50
> >
> > | Dataset | Methods | $\alpha$ | Accuracy | ECE | Brier | NLL |
> > | --- | --- | :---: | :---: | :---: | :---: | :---: |
> > |  C10 | ERM Baseline | -- | 94.26 $\pm$ 0.12 | 0.56 $\pm$ 0.05 | 8.56 $\pm$ 0.23 | 17.93 $\pm$ 0.36
> > |  | Mixup | 1 | 95.6 $\pm$ 0.17 | 1.40 $\pm$ 0.12 | 7.13 $\pm$ 0.31 | 17.32 $\pm$ 0.88
> > |  |  | 0.5 | 95.53 $\pm$ 0.18 | 1.29 $\pm$ 0.15 | 7.22 $\pm$ 0.30 | 17.44 $\pm$ 0.66
> > |  |  | 0.1 | 94.98 $\pm$ 0.25 | 1.29 $\pm$ 0.21 | 7.83 $\pm$ 0.37 | 17.84 $\pm$ 0.78
> > |  | Mixup IO | 1 | 94.74 $\pm$ 0.34 | 0.47 $\pm$ 0.07 | 7.78 $\pm$ 0.41 | 16.13 $\pm$ 0.75
> > |  |  | 0.5 | 95.07 $\pm$ 0.17 | 0.48 $\pm$ 0.08 | 7.39 $\pm$ 0.14 | 15.23 $\pm$ 0.34
> > |  |  | 0.1 | 94.79 $\pm$ 0.06 | 0.7 $\pm$ 0.16 | 7.85 $\pm$ 0.20 | 16.37 $\pm$ 0.61
> > |  | Manifold Mixup | 2 | 95.86 $\pm$ 0.18 | 1.04 $\pm$ 0.23 | 6.93 $\pm$ 0.27 | 17.18 $\pm$ 0.66
> > |  |  | 1 | 96.02 $\pm$ 0.08 | 1.32 $\pm$ 0.35 | 6.64 $\pm$ 0.15 | 16.72 $\pm$ 0.34
> > |  |  | 0.5 | 95.64 $\pm$ 0.31 | 1.36 $\pm$ 0.11 | 7.15 $\pm$ 0.51 | 17.63 $\pm$ 1.11
> > |  |  | 0.1 | 94.79 $\pm$ 0.34 | 1.19 $\pm$ 0.16 | 8.4 $\pm$ 0.54 | 19.77 $\pm$ 1.45
> > |  | RegMixup | 20 | 96.14 $\pm$ 0.15 | 0.91 $\pm$ 0.06 | 6.41 $\pm$ 0.23 | 14.77 $\pm$ 0.33
> > |  | MIT-A ($\Delta \lambda > 0.5$) | 1 | 95.68 $\pm$ 0.28 | 0.88 $\pm$ 0.19 | 6.58 $\pm$ 0.43 | 13.88 $\pm$ 0.83
> > |  | MIT-L ($\Delta \lambda > 0.5$) | 1 | 95.42 $\pm$ 0.14 | 0.66 $\pm$ 0.08 | 6.85 $\pm$ 0.18 | 14.41 $\pm$ 0.32
> > |  | Kernel Warping Mixup (Ours) | 1 | 95.82 $\pm$ 0.04 | 0.65 $\pm$ 0.09 | 6.86 $\pm$ 0.05 | 17.05 $\pm$ 0.3
> > |  C100 | ERM Baseline | -- | 73.83 $\pm$ 0.82 | 2.20 $\pm$ 0.13 | 35.90 $\pm$ 1.04 | 96.39 $\pm$ 3.45
> > |  | Mixup | 1 | 78.05 $\pm$ 0.23 | 2.41 $\pm$ 0.23 | 31.26 $\pm$ 0.26 | 88.01 $\pm$ 0.53
> > |  |  | 0.5 | 78.51 $\pm$ 0.37 | 2.55 $\pm$ 0.22 | 30.44 $\pm$ 0.44 | 85.57 $\pm$ 1.88
> > |  |  | 0.1 | 76.49 $\pm$ 0.86 | 2.69 $\pm$ 0.13 | 32.75 $\pm$ 1.05 | 89.82 $\pm$ 3.87
> > |  | Mixup IO | 1 | 75.25 $\pm$ 0.72 | 1.77 $\pm$ 0.13 | 34.24 $\pm$ 0.68 | 91.41 $\pm$ 2.18
> > |  |  | 0.5 | 76.42 $\pm$ 0.81 | 1.94 $\pm$ 0.15 | 32.65 $\pm$ 1.01 | 86.1 $\pm$ 3.04
> > |  |  | 0.1 | 75.82 $\pm$ 0.98 | 2.1 $\pm$ 0.22 | 33.45 $\pm$ 1.26 | 89.54 $\pm$ 3.75
> > |  | Manifold Mixup | 2 | 81.54 $\pm$ 0.12 | 2.2 $\pm$ 0.14 | 26.94 $\pm$ 0.15 | 73.15 $\pm$ 0.75
> > |  |  | 1 | 80.39 $\pm$ 0.31 | 2.58 $\pm$ 0.07 | 28.54 $\pm$ 0.34 | 79.06 $\pm$ 0.96
> > |  |  | 0.5 | 79.46 $\pm$ 0.91 | 2.76 $\pm$ 0.30 | 29.63 $\pm$ 1.09 | 82.92 $\pm$ 3.43
> > |  |  | 0.1 | 76.85 $\pm$ 1.28 | 2.87 $\pm$ 0.28 | 32.54 $\pm$ 1.49 | 90.09 $\pm$ 4.92
> > |  | RegMixup | 10 | 78.44 $\pm$ 0.24 | 2.20 $\pm$ 0.23 | 30.82 $\pm$ 0.29 | 83.16 $\pm$ 1.19
> > |  | MIT-A ($\Delta \lambda > 0.5$) | 1 | 77.81 $\pm$ 0.42 | 2.19 $\pm$ 0.05 | 30.84 $\pm$ 0.53 | 80.49 $\pm$ 1.45
> > |  | MIT-L ($\Delta \lambda > 0.5$) | 1 | 77.14 $\pm$ 0.71 | 2.13 $\pm$ 0.17 | 31.74 $\pm$ 1.11 | 82.87 $\pm$ 3.24
> > |  | Kernel Warping Mixup (Ours) | 1 | 79.62 $\pm$ 0.68 | 1.84 $\pm$ 0.22 | 29.18 $\pm$ 0.78 | 80.46 $\pm$ 2.08
> >
> > ### Table 3: Performance and calibration comparison with Resnet50 on Tiny-Imagenet
> >
> > | Method | $\alpha$ | Accuracy | ECE | Brier | NLL |
> > |---|:---:|:---:|:---:|:---:|:---:|
> > | ERM Baseline | - | 66.74 $\pm$ 0.34 | 1.62 $\pm$ 0.22 | 1.36 $\pm$ 0.02 | 0.44 $\pm$ 0.0045 |
> > | Mixup | 1 | 67.21 $\pm$ 0.21 | 1.63 $\pm$ 0.10 | 1.37 $\pm$ 0.01 | 0.44 $\pm$ 0.0028 |
> > | | 0.5 | 67.34 $\pm$ 0.69 | 1.56 $\pm$ 0.05 | 1.36 $\pm$ 0.04 | 0.44 $\pm$ 0.0099 |
> > | | 0.1 | 66.48 $\pm$ 0.57 | 1.66 $\pm$ 0.17 | 1.42 $\pm$ 0.02 | 0.45 $\pm$ 0.0061 |
> > | MixupIO | 1 | 66.17 $\pm$ 0.28 | 1.49 $\pm$ 0.21 | 1.37 $\pm$ 0.01 | 0.45 $\pm$ 0.0033 |
> > | | 0.5 | 66.98 $\pm$ 0.39 | 1.75 $\pm$ 0.12 | 1.35 $\pm$ 0.01 | 0.44 $\pm$ 0.0025 |
> > | | 0.1 | 65.87 $\pm$ 0.57 | 1.51 $\pm$ 0.21 | 1.39 $\pm$ 0.02 | 0.45 $\pm$ 0.0057 |
> > | RegMixup | 10 | 69.39 $\pm$ 0.62 | 1.31 $\pm$ 0.08 | 1.27 $\pm$ 0.03 | 0.42 $\pm$ 0.0078 |
> > | RegMixup | 20 | 69.71 $\pm$ 0.42 | 1.17 $\pm$ 0.19 | 1.24 $\pm$ 0.03 | 0.41 $\pm$ 0.0067 |
> > | Kernel Warping Mixup (Ours) | 1 | 68.18 $\pm$ 0.26 | 1.29 $\pm$ 0.35 | 1.33 $\pm$ 0.02 | 0.43 $\pm$ 0.0046 |

---

### Meta-Review · Area_Chair_DEpT · 2023-12-05

**Metareview:**

The reviewers found this paper interesting and aspects of the approach and evaluation setup novel. However, all reviewers expressed several concerns related to the relevance of the problem setup for mixup, bold claims about novelty, and different aspects in the experimental validation of the approach. The authors did a good job in the rebuttal to sort out several concerns, but overall, some of the main concerns related to to motivation/insight of the proposed method and showing the benefits in wide enough empirical validation remain. The scores are mixed and the paper is borderline. It would do this paper good to go through a proper revision and be resubmitted to the next venue.

**Justification For Why Not Higher Score:**

I would not be upset to see this paper accepted, but I don't think it reaches the bar for acceptance in its current form.

**Justification For Why Not Lower Score:**

N/A

---

### Decision · Program_Chairs · 2024-01-16

Reject